# A *Toxoplasma gondii* O-glycosyltransferase that modulates bradyzoite cyst wall rigidity is distinct from host homologues

Pranav Kumar[1], Tadakimi Tomita[2], Thomas A. Gerken [3], Collin J. Ballard[3], Yong Sok Lee[4], Louis M. Weiss[2,5] & Nadine L. Samara [1]✉

Infection with the apicomplexan protozoan *Toxoplasma gondii* can be life-threatening in immunocompromised hosts. Transmission frequently occurs through the oral ingestion of *T. gondii* bradyzoite cysts, which transition to tachyzoites, disseminate, and then form cysts containing bradyzoites in the central nervous system, resulting in latent infection. Encapsulation of brady-zoites by a cyst wall is critical for immune evasion, survival, and transmission. O-glycosylation of the protein CST1 by the mucin-type O-glycosyltransferase *T. gondii* (Txg) GalNAc-T3 influences cyst wall rigidity and stability. Here, we report X-ray crystal structures of TxgGalNAc-T3, revealing multiple features that are strictly conserved among its apicomplexan homologues. This includes a unique 2nd metal that is coupled to substrate binding and enzymatic activity in vitro and cyst wall O-glycosylation in *T. gondii*. The study illustrates the divergence of pathogenic protozoan GalNAc-Ts from their host homologues and lays the groundwork for studying apicomplexan GalNAc-Ts as therapeutic targets in disease.

*Toxoplasma gondii* is an obligate intracellular protozoan pathogen belonging to the phylum Apicomplexa that infects a wide range of warm-blooded animals, including humans[1]. It is transmitted through the oral ingestion of bradyzoites found in tissue cysts (found in raw or undercooked meat) or by the ingestion of sporozoites residing in oocysts (found in feline feces) contaminating food or water[2]. *T. gondii* tissue cysts reside in in the central nervous system or muscle tissue and are associated with latent infection. During recrudescence, brady-zoites within tissue cysts convert to tachyzoites and disseminate resulting in infections that can be life threatening in immunocom-promised hosts such as HIV/AIDS patients[3]. While there are drugs that target tachyzoites, the active replicating life cycle state, *T. gondii* tissue cysts have thus far been resistant to treatment[4]. Thus, an effective

medication that targets bradyzoites in tissue cysts is required for preventing reactivation and acute infection.

The *T. gondii* tissue cyst is a modified parasitophorous vacuole within a host cell that is surrounded by a wall that is crucial for the survival and transmission of *T. gondii*. The cyst wall contains a subset of proteins that that pass through the secretory pathway and undergo mucin-type O-glycosylation, a post-translational modification (PTM) that is conserved across higher eukaryotes and a subset of apicom-plexan protozoa and influences protein structure and function[5,6]. *T. gondii* O-glycosylated cyst wall proteins include CST1 (TGME49_264660)[7], SRS13 (TGME49_222370)[8], proteophosphoglycan PPG1 (TGME49_297520)[7,9], and GRA2 (TGME49_227620)[10]. CST1 is an abundant protein that contains thirteen SAG1 related sequence (SRS)

[1]Structural Biochemistry Unit, National Institute of Dental and Craniofacial Research, NIH, Bethesda, MD 20892, USA. [2]Department of Pathology, Albert Einstein College of Medicine, Bronx, 1300 Morris Park Avenue, New York 10461, USA. [3]Departments of Biochemistry and Chemistry, Case Western Reserve University, Cleveland, OH 44106, USA. [4]Bioinformatics and Computational Biosciences Branch, National Institute of Allergy and Infectious Diseases, NIH, Bethesda, MD 20892, USA. [5]Department of Medicine (Infectious Disease), Albert Einstein College of Medicine, Bronx 1300 Morris Park Avenue, New York 10461, USA. ✉e-mail: nadine.samara@nih.gov

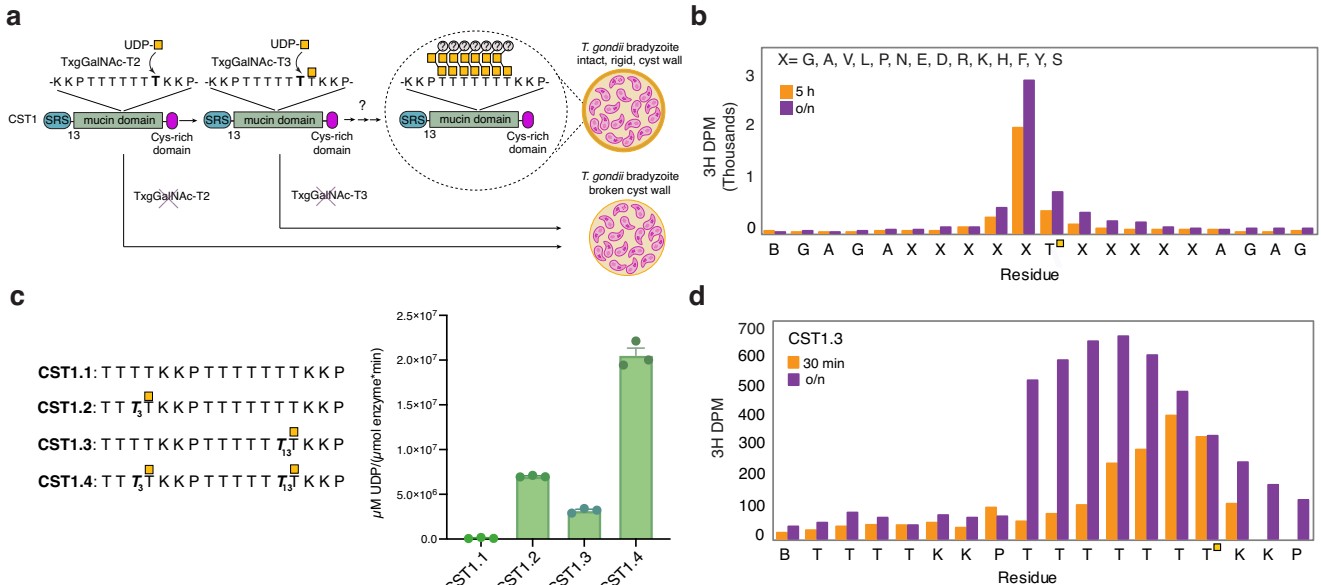

**Fig. 1 | Mucin-type O-glycosylation by TxgGalNAc-T3. a** O-glycosylation of the mucin domain (green, containing 20 threonine rich repeats, with one repeat shown here) of the cyst wall protein CST1 by TxgGalNAc-T2 and TxgGalNAc-T3 occurs in a hierarchal manner. TxgGalNAc-T2 initiates the process by O-glycosylating a unglycosylated acceptor Thr, and TxgGalNAc-T3 recognizes a Thr-O-GalNAc at the +1 position to glycosylate adjacent sites in the N-terminal direction. Chain extension is predicted to result in the addition of a second GalNAc, followed by the transfer of an unknown capping sugar, but the mechanism of these processes and the enzymes involved are not known. O-glycosylation of CST1 influences cyst wall rigidity and knocking out each transferase results in a fragile cyst wall (shown in dark yellow). **b** Edman degradation assay that directly measures GalNAc transfer to a peptide using the GPIIC random library shows that TxgGalNAc-T3 specificity is dictated by GalNAc at the +1 position in a sequence independent manner. **c** Indirect measurements of TxgGalNAc-T3 O-glycosylation of CST1 glycopeptides using a UDP-Glo assay to quantify sugar nucleotide hydrolysis. The data shows an increase in activity with additional Thr-O-GalNAcs on the CST1 peptide substrates, where $n = 3$ biologically independent experiments performed in triplicate each time (9 technical replicates). The error bar represents the standard error of the mean (SEM). **d** Edman degradation assay showing that O-glycosylation by TxgGalNAc-T3 is distributive after 30 min, with decreasing efficiency after each addition. TxgGalNAc-T3 densely glycosylates the CST1.3 mono-glycopeptide after an overnight reaction. (Note that the 3H DMP observed at and C-terminal of Thr14 is due to Edman sequencer lag of the 3H-GalNAc-O-Thr-PTH derivative). Source data are provided as a Source Data file.

domains found in a superfamily of *T. gondii* surface antigens that could facilitate parasite entry into host cells[11], an O-glycosylated mucin domain with 20 threonine-rich repeats, and a C-terminal cysteine-rich domain (Fig. 1a)[7]. The deletion of CST1 or its mucin domain reduces bradyzoite growth rate and cyst formation, results in dysregulated bradyzoite gene expression, and produces cysts that are fragile in comparison to intact wild-type cysts, suggesting that the mucin domain is critical for both the formation of an organized and structurally stable cyst wall and bradyzoite persistence[7].

O-glycosylation of the CST1 mucin domain is initiated by GalNAc-Ts, a family of Golgi membrane-anchored enzymes that initiate mucin-type O-glycosylation by catalyzing the transfer of N-acetylgalactosamine (GalNAc) from UDP-GalNAc to Thr/Ser to form Thr/Ser-O-GalNAc[12,13]. GalNAc-Ts consist of a luminal glycosyltransferase family A (GT-A) type catalytic domain (CAZy family GT27) with a Rossmann-like fold and a conserved active site DXH motif, where Asp and His help coordinate an essential catalytic $Mn^{2+}$[14]. A short linker (~10 aa) connects the catalytic domain to a C-terminal Ricin B-type lectin domain that, depending on the isoenzyme, has the potential to interact with existing Thr/Ser-O-GalNAc on substrates or through direct contacts with the peptide backbone to influence substrate specificity[15–20].

*T. gondii* has five GalNAc-Ts[21–23] (Fig. S1a): TxgGalNAc-T1, TxgGalNAc-T2, and TxgGalNAc-T3 are expressed in the bradyzoite and tachyzoite stages[24]. TxgGalNAc-T4 and TxgGalNAc-T5 are expressed in the oocyst stage[24,25]. The role of mucin-type O-glycosylation in tachyzoites and oocysts is currently not known. In bradyzoites, TxgGalNAc-T2 initiates the O-glycosylation of the Thr-rich mucin domain of CST1 by modifying unglycosylated regions. The strict glycopeptide-preferring enzyme TxgGalNAc-T3 then adds GalNAc to acceptor Thr one position N-terminal to existing Thr-O-GalNAcs and is predicted to

sequentially O-glycosylate Thr rich repeats within the mucin domain of CST1. The extension of each initial GalNAc could form a core-5 O-glycan (GalNAc-1,3GalNAc 1-Thr) with an unknown capping sugar, although little is known about this process (Fig. 1a)[24,26]. Knocking out TxgGalNAc-T2 or TxgGalNAc-T3 results in cyst wall breakage, demonstrating that O-glycosylation is critical for generating a rigid cyst wall (Fig. 1a)[24].

A low sequence identity between TxgGalNAc-T3 and the 20 human isoenzymes hints at an evolutionary divergence in substrate recognition and enzyme function (Fig. S1b). To gain insight into the differences between this enzyme and its human homologs, we solved x-ray crystal structures of TxgGalNAc-T3 alone or in complex with glycopeptide substrates. The structures reveal a unique GalNAc binding pocket that dictates the substrate preference of TxgGalNAc-T3, which is coupled to a 2nd metal binding site that is strictly conserved among apicomplexan homologs. We also identify an active site Glu that is only present in apicomplexan homologs of TxgGalNAc-T3 and is oriented towards the acceptor Thr, suggesting a distinct catalytic mechanism from human isoenzymes. In addition, TxgGalNAc-T3 has an extended C-terminal tail that is critical for its function, and a second substrate binding loop that does not similarly bind substrates in human GalNAc-Ts. Our biochemical, computational, and in vivo *T. gondii* cyst wall glycosylation data support the structural observations and demonstrate how these apicomplexa-specific characteristics are critical for the enzymatic function of TxgGalNAc-T3.

## Results
### Substrate specificity of TxgGalNAc-T3
We purified the luminal region (catalytic and lectin domains, aa 74–635) of TxgGalNAc-T3 (AY160970.1) after secreted expression in

*Pichia pastoris.* Previous studies using various peptides showed that TxgGalNAc-T3 does not modify unglycosylated peptides but glycosylates one residue N-terminal to Thr-O-GalNAc on a glycopeptide (Thr-O-GalNAc at the +1 position)[22]. We confirmed this preference using the Muc5AC mucin (glyco)peptide series (Fig. S2a). The data show that TxgGalNAc-T3 does not glycosylate the nascent Muc5AC peptide, but readily glycosylates the mono-glycopeptides Muc5AC-3 and Muc5AC-13 and the di-glycopeptide Muc5AC-3,13 (Fig. S2a). To assess the influence of the surrounding amino acid sequence on substrate specificity, we used a random glycopeptide substrate library (GPIIC)[20] to demonstrate that the activity enhancement of TxgGalNAc-T3 by Thr-O-GalNAc at the +1 position of an acceptor site is sequence independent (−5 to +5 aa, Fig. 1b).

We then verified the substrate preference of TxgGalNAc-T3 for endogenous *T. gondii* cyst wall proteins using peptides and glycopeptides from CST1 and SRS13. For CST1, TxgGalNAc-T3 does not efficiently glycosylate the unmodified peptide CST1.1, is similarly active towards the mono-glycopeptides CST1.2 and CST1.3 and has the highest activity against the di-glycopeptide CST1.4, as expected (Fig. 1c). TxgGalNAc-T3 does not readily modify the unglycosylated SRS13.1 peptide, as predicted (Fig. S2b). Surprisingly, there is diminished activity for the C-terminal mono-glycopeptide SRS13.3, while activity for the SRS13.4 di-glycopeptide is ~1.5-fold lower than the SRS13.2 mono-glycopeptide despite the availability of two potential acceptor sites, suggesting that either the C-terminal acceptor site or proximal amino acids are inhibiting enzymatic activity. Overall, the differences in activity among the various peptide series suggests that while amino acid residues proximal to Thr-O-GalNAc in a substrate do not affect the +1 preference of TxgGalNAc-T3, they can influence its O-glycosylation efficiency.

## TxgGalNAc-T3 sequentially glycosylates by a distributive mechanism

It was previously shown that the deletion of TxgGalNAc-T3 in *T. gondii* results in the disappearance of high molecular weight (Mw), but not intermediate Mw CST1. Rescue with TxgGalNAc-T3 restores the high Mw species, supporting the hierarchal model in which TxgGalNAc-T2 initially glycosylates CST1 to produce Intermediate bands, followed by TxgGalNAc-T3 O-GalNAc mediated recognition of CST1 to glycosylate the remaining sites[24]. We thus assessed the ability of TxgGalNAc-T3 to glycosylate every putative acceptor site on our glycopeptide substrates. For the CST1 glycopeptides, TxgGalNAc-T3 sequentially glycosylates sites N-terminal to the Thr-O-GalNAc with decreasing efficiency upon each GalNAc addition in 30- or 90-min reactions (as demonstrated in Figs. 1d, S2c, d) and can eventually glycosylate all six N-terminal Thr on CST1.3 in an overnight reaction (Figs. 1d and S2e). We observe similar sequential and gradually decreasing glycosylation efficiency for the Muc5AC glycopeptides and SRS13.2 after 90 mins (Fig. S2f–i).

The decreasing efficiency upon each GalNAc addition suggests that TxgGalNAc-T3 O-glycosylates its substrates by a distributive mechanism, where substrate release occurs after a single GalNAc is added and TxgGalNAc-T3 forms a unique enzyme-substrate complex for each addition, with decreasing affinity for the substrate with each additional GalNAc[27]. Interestingly, TxgGalNAc-T3 can efficiently glycosylate Thr then Ser on Muc5AC-13, followed by a striking decrease in the efficiency of glycosylation at the sites N-terminal to Ser, suggesting that TxgGalNAc-T3 specifically prefers Thr-O-GalNAc over Ser-O-GalNAc at the +1 position (Fig. S2h). Indeed, changing Thr-O-GalNAc to Ser-O-GalNAc on a glycopeptide results in a 5-fold increase in the $K_M$, demonstrating diminished binding to Ser-O-GalNAc (Fig. S2j). Nevertheless, after an overnight incubation, TxgGalNAc-T3 will glycosylate all four acceptors in the TTSTT* sequence of Muc5AC-13 (Fig. S2k).

## X-ray crystal structures of TxgGalNAc-T3 bound to substrates

To investigate the basis of the substrate specificity and function of TxgGalNAc-T3, we solved the x-ray co-crystal structures of TxgGalNAc-T3 in complex with $Mn^{2+}$, a non-hydrolysable form of UDP-GalNAc (UDP-2-(acetylamino)−4-F-D-galactosamine disodium salt, UDP-GalNAc-F) and each of the following glycopeptides: CST1.4, Muc5AC-3,13, Muc5AC-3, Muc5AC-13, and SRS13.2 from 2.2-2.9 Å resolution (Figs. 2a, S3a–d, Table S1). The structures reveal a similar architecture to their metazoan homologs (Fig. S4a). The catalytic domain adopts the conserved GT-A fold and the active site contains a DXH motif (Asp276, Ser277, His278), where Asp276, His278, His414, UDP, and a water molecule coordinate $Mn^{2+}_A$ with octahedral geometry (Fig. 2b)[16,17,19,28–32]. An RMSD comparison of the TxgGalNAc-T3 structure with metazoan homologs shows a structurally similar and superposable catalytic domain, whereas the lectin domain orientations in the structures vary (and this is also true when metazoan homologs are compared to each other, Fig. S4b)[17–19,28,30,31,33]. An overlay of sequence similarity between TxgGalNAc-T3 and the human isoenzymes over the TxgGalNAc-T3 structure shows that the core of the catalytic domain is most conserved, whereas the peripheral regions are less conserved, and the lectin domain is the least conserved among the isoenzymes. TxgGalNAc-T3 also contains regions that are absent in the human isoenzymes (Fig. S4c).

We modeled UDP into the active site but did not include GalNAc-F due to weak electron density, either due to the high mobility of the GalNAc moiety or hydrolysis of UDP-GalNAc-F during crystallization (Fig. 2b). Sufficient electron density allowed us to model the N-terminal residues (in brackets) of the di-glycopeptides CST1.4 ([TTT₃T^GalNAcKKP]TTTTTTT^GalNAcKKP, Fig. 2a) and Muc5AC-3,13 ([GT₂T^GalNAcPSP]VPTTSTT^GalNAcSAP, Fig. S3a) that include the acceptor Thr (Thr3 for CST1.4 and Thr2 for Muc5AC-3,13) and Thr-O-GalNAc. The peptides were refined with similar B-factors to proximal residues in TxgGalNAc-T3 (Fig. S5a, b). We further verified the peptide density by calculating unbiased omit maps (Fig. S5c, d). Although the glycopeptide amino acid sequences vary, both CST1.4 and Muc5AC-3,13 are similarly bound along the active site substrate groove with the acceptor Thr positioned for catalysis, adopting a similar conformation seen in peptides bound to metazoan GalNAc-Ts (Figs. 2a, S5e)[17,19,29,33].

Poor electron density for the remaining unmodeled residues in CST1.4 and Muc5AC-3,13 suggests that they are disordered in the structure and making transient interactions with TxgGalNAc-T3. For the mono-glycopeptides, we observe density for Thr-O-GalNAc in the TxgGalNAc-T3:Muc5AC-3 co-crystal structure (Fig. S3b), but disconnected and weak density for the peptide residues. We could not model the peptide substrate in the structure containing Muc5AC-13 due to unresolvable electron density, and we only observe weak density for Thr-O-GalNAc in the SRS13.2 structure (Fig. S3c, d). It is notable that the di-glycopeptides were the only substrates with electron density for the N-terminal portion of the peptide chain, but it is not clear what is stabilizing the peptide conformations in these structures since we did not observe density for the C-terminal residues. Overall, the lack of density for the peptide backbone or Thr-O-GalNAc at the C-terminal ends of the glycopeptides suggests a lack of strong interactions with the catalytic or lectin domain of TxgGalNAc-T3.

We then assessed the basis of the TxgGalNAc-T3 substrate preference. Thr-O-GalNAc at the +1 position on the glycopeptides is bound to a pocket in TxgGalNAc-T3 consisting of His333 (catalytic domain) and Glu554 (lectin domain) that interact with GalNAc via sidechain H-bonds, and Ser334 (catalytic domain), which interacts with GalNAc through a mainchain H-bond (Figs. 2b, S6a). We observed density that appears to be a $2^{nd}$ metal ($Mn^{2+}_N$) near the GalNAc binding pocket, a striking and unexpected feature not previously seen in metazoan GalNAc-Ts (Fig. 2b). $Mn^{2+}_N$ is coupled to the GalNAc binding pocket through coordination to His333 and indirect coordination to Glu554

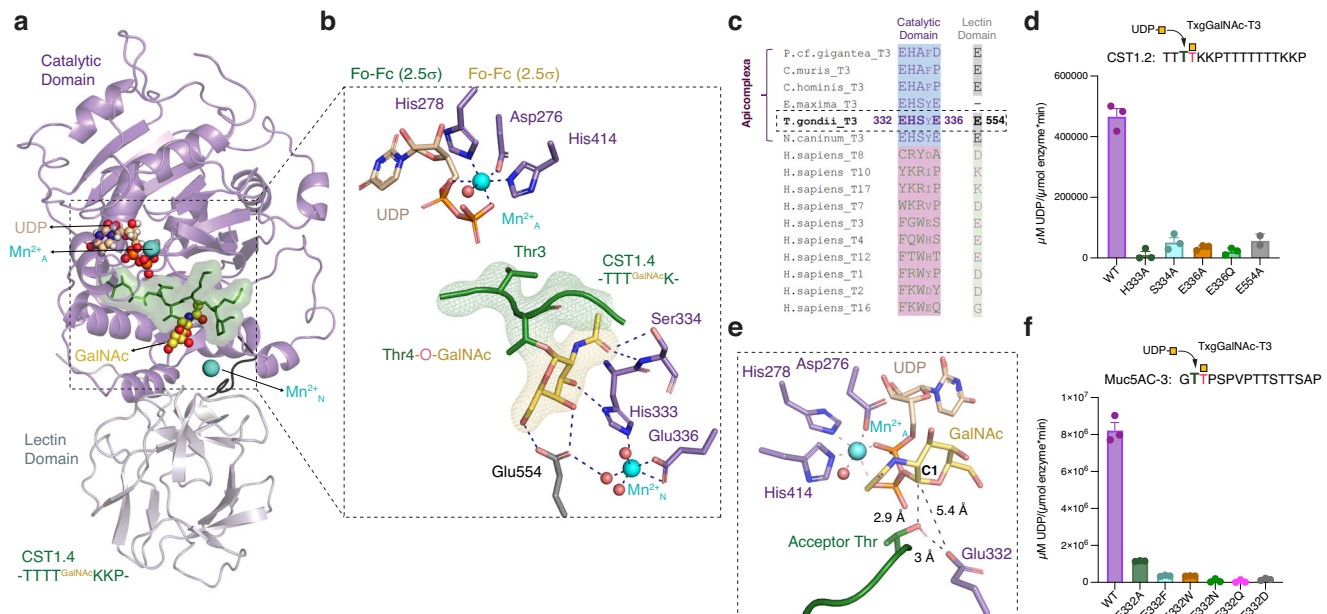

**Fig. 2 | X-ray structure of TxgGalNAc-T3, UDP, Mn²⁺, and a CST1 di-glycopeptide. a** TxgGalNAc-T3, shown with the catalytic domain in lavender and the C-terminal lectin domain in grey. The CST1.4 di-glycopeptide is green, GalNAc is yellow, Mn²⁺ is aquamarine, and UDP is wheat. **b** The active site adopts the conserved GT-A fold with a $D_{276}XH_{278}$ metal-binding (Mn²⁺_A) motif. The acceptor Thr3 is positioned for GalNAc transfer. GalNAc on the +1 Thr4-O-GalNAc is bound to a pocket that is coupled to a 2ⁿᵈ metal binding site (Mn²⁺_N). A CST1.4 Fo-Fc omit map is shown in green and the GalNAc omit map is yellow, both contoured at 2.5σ. **c** Sequence alignment comparing the GalNAc binding pocket residues of apicomplexan homologs (blue highlight) of TxgGalNAc-T3 to a subset of representative human homologs (pink highlight) showing the site is not conserved in higher eukaryotes. **d** UDP-Glo activity assay comparing TxgGalNAc-T3^WT to variants with disruptions in the GalNAc binding pocket and 2ⁿᵈ metal binding pocket showing that this site is important for enzymatic function using CST1.2 as a substrate. **e** The residue Glu332 is positioned near the acceptor Thr. **f** UDP-Glo activity assay showing that Glu332 influences enzymatic activity using Muc5AC-3 as a substrate. For each variant in **d,f**, n = 3 biologically independent experiments performed in triplicate each time (9 technical replicates). The error bars in **d,f** represent the standard error of the mean (SEM). Source data are provided as a Source Data file.

via water. Glu336 and 2 additional waters complete the coordination sphere of Mn²⁺_N, which adopts an irregular octahedral geometry (Fig. 2b). The GalNAc and Mn²⁺_N binding residues are conserved in most apicomplexan homologs of TxgGalNAc-T3, but diverge from the corresponding region in human GalNAc-Ts, including GalNAc-T10, T7, and T17, which also strictly recognize GalNAc at the +1 position and modify the adjacent acceptor site (Fig. 2c)[20,34].

We evaluated the influence of the Thr-O-GalNAc and Mn²⁺_N binding residues on activity by producing His333, Ser334, Glu554, and Glu336 variants using site directed mutagenesis and assessed variant stability by size exclusion chromatography and Nano Differential Scanning Fluorimetry (Nano-DSF) (Fig. S6b). We obtained varying yields for the variants (Table S2), but the melting temperatures (T_M) were within 4 °C of the TxgGalNAc-T3^WT T_M. We considered the possibility that Mn²⁺_N binding is an artefact of crystals grown or soaked in high concentrations of Mn²⁺ and made two variants of Glu336 (E336A and E336Q), the only residue in the motif that coordinates Mn²⁺_N but does not interact with GalNAc. We predicted that any effects on activity for E336A and E336Q are directly attributed to Mn²⁺_N binding. We then measured activity for all variants and show that changes to the GalNAc pocket and Mn²⁺_N binding residues diminish activity compared to TxgGalNAc-T3^WT (Fig. 2d). The data verify that the Thr-O-GalNAc binding pocket residues influence TxgGalNAc-T3 activity and demonstrate a critical role for Mn²⁺_N in enzymatic function.

Interestingly, Glu332 is positioned ~3 Å from the acceptor Thr and ~5.5 Å from the anomeric C1 carbon on GalNAc when we model UDP-GalNAc into the structure using the available density (Fig. 2e). In human isoenzymes, the residue at that position is frequently Arg, Trp, or Tyr (Fig. 2c). Changes in Glu332 to Ala or any other metazoan residues abrogates activity. Additionally, residues with polar side chains or Asp also result in diminished activity, suggesting a unique key role for

Glu332 in catalysis (Fig. 2f). To understand how Glu332 influences activity, we performed enzyme kinetics on TxgGalNAc-T3^WT and TxgGalNAc-T3^E332A using Muc5AC-3 as a substrate. A ~16-fold decrease in k_cat and ~5-fold decrease in K_M for TxgGalNAc-T3^E332A compared to TxgGalNAc-T3^WT indicates that Glu332 primarily influences reaction chemistry (Fig. S7a, b, Table S3). Glu332 appears to align and deprotonate the acceptor Thr. Alternatively, Glu332 could be a nucleophile in a double displacement reaction if conformational changes during catalysis brings it closer to C1 on GalNAc (Fig. 2e)[14].

## Assessing the role of Mn²⁺_N in catalysis

To examine the role of Mn²⁺_N in catalysis, we first verified its presence by solving the x-ray crystal structure of apo TxgGalNAc-T3 in the absence of Mn²⁺ to 2.9 Å resolution and did not observe density for Mn²⁺_N (Fig. 3a, Table S4). We then solved the structure to 2.5 Å resolution after soaking the apo crystals in Mn²⁺ and detected the appearance of electron density and an anomalous signal for Mn²⁺_N (Fig. 3b, Table S4). We initially hypothesized that Mn²⁺_N was aligning the sidechains for GalNAc binding. However, a comparison of the substrate-bound structure and apo structure shows that the Mn²⁺_N binding residues adopt similar conformations in both structures, indicating that Mn²⁺_N uses an alternate mechanism to influence activity (Fig. 3c).

We reasoned that the positive charge on Mn²⁺_N influences GalNAc binding and reaction chemistry. To assess the effect of electrostatics on activity, we measured enzymatic activity at various pHs. Optimal activity occurs between pH 7.3-8.2 and decreases as pH increases (Fig. S7c). However, the crystals form at pH 9.5, where activity is low in solution. Indeed, kinetics show that at pH 9.0, the k_cat is ~185-fold less than at pH 7.3 and the K_M is ~2-fold less, suggesting tighter substrate binding but lower turnover (Fig. S7d). At pH 9-9.5, the acidic side

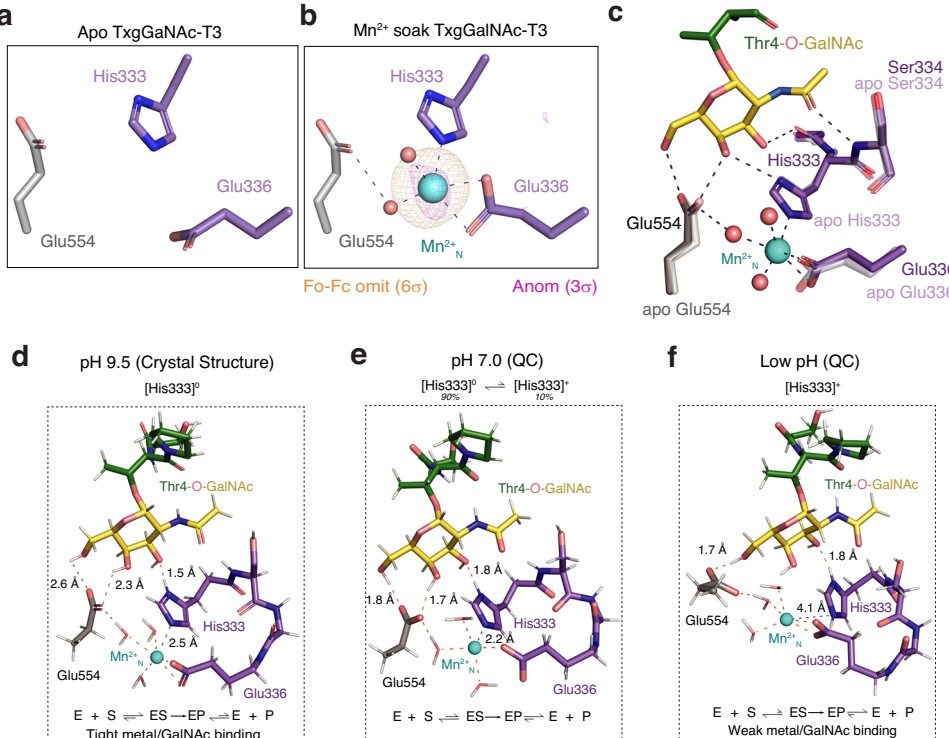

**Fig. 3 | Role of the second metal binding site (Mn²⁺$_N$) in catalysis. a** Electron density for Mn²⁺$_N$ is absent in crystals grown in the absence of Mn²⁺. **b** Soaking apo-TxgGalNAc-T3 crystals in Mn²⁺ results in the appearance of Mn²⁺$_N$, where the Fo-Fc omit map is contoured at 6σ (orange) and the anomalous signal (data collection λ =1 Å) is contoured at 3σ (pink). **c** There are no differences in the sidechain conformations of the metal binding residues in the apo structure (no Mn²⁺) and the co-crystal complex containing di-glycopeptide substrate. **d–f** The second metal influences catalysis by regulating Michaelis complex formation (E:S) in a pH sensitive manner. **d** Conformation of the site in the crystal structure at pH 9.5.

Deprotonated sidechains bind tightly to Mn²⁺ and GalNAc, forming a tight Michaelis complex and inhibiting product release. **e** Quantum chemistry calculations at pH 7.0, where His deprotonation promotes ES complex formation, and His protonation promotes product release. **f** Quantum chemistry calculations where the His333 sidechain is fully protonated. Here, His333 is no longer binding Mn²⁺, and Glu554 interactions with GalNAc are reduced, disrupting ES complex formation and catalysis. Quantum chemistry generated coordinate files are included as Supplementary Dataset 1.

chains are completely deprotonated and His333 is neutral and can coordinate Mn²⁺. Thus, the state observed in the crystal at pH 9.5 represents a possible stable enzyme:substrate (E:S) complex where substrate is tightly bound, and catalytic activity and turnover are reduced, since GalNAc-Ts function optimally at around neutral pH (Fig. 3d)[35]. We could not obtain diffracting crystals at neutral pH, so we soaked crystals grown at pH 9.5 into a pH 7.3 buffer before freezing and did not observe bound Muc5AC-3,13 di-glycopeptide or Mn²⁺$_N$ in the structure (Table S4).

We then utilized quantum chemistry calculations to assess the effect of the protonation state of His333 on the local geometry around Mn²⁺$_N$. The geometry-optimized structure with neutral His333 depicts it as strongly bonded to Mn²⁺$_N$ as indicated by the bond distance of ~2.2 Å, while Glu554 is also H-bonded to two hydroxyls of GalNAc (Fig. 3e). At an artificially low pH with 100% [His333]⁺, such strong interactions are not seen (Fig. 3f). His333 does not coordinate Mn²⁺$_N$ and Glu554 H-bonds with one hydroxyl of GalNAc, suggesting weaker binding to the glycopeptide substrate (Fig. 3f). The difference in Mn²⁺$_N$ and glycopeptide binding at varying pHs may explain why the peptide ligands dissociate easily from the binding pocket when His333 is protonated. At pH 7, His is at an equilibrium between uncharged (~90%) and charged (~10%) species. Thus, the E:S complex can readily form when His333 is uncharged and coordinating Mn²⁺$_N$, catalysis can occur, and the product is released due to His333 protonation (Fig. 3e).

To test this model, we changed His333 to Asn (H333N), since Asn can interact with GalNAc, but does not effectively coordinate Mn²⁺$_N$, therefore decoupling Mn²⁺$_N$ and GalNAc binding and mimicking His333 binding to Mn²⁺$_N$ at low pH. TxgGalNAc-T3^H333N results in

a ~17-fold decrease in the $K_M$ and ~100-fold decrease in $k_{cat}$ compared to TxgGalNAc-T3^WT, indicating that Mn²⁺$_N$ coordination is fine-tuning substrate binding and catalysis in a pH-dependent manner (Fig. S7e and Table S3). Since Glu332 is adjacent to His333 on a loop, metal binding could both affect its pKa and influence loop flexibility in a manner that allows Glu332 to engage in a nucleophilic attack on GalNAc-C1 in a double displacement mechanism: when metal binding is tight (high pH), Glu332 is restricted, and when metal binding is weaker (low pH), the loop is flexible allowing Glu332 to approach C1. Finally, we show that the presence of Mn²⁺$_N$ does not influence the overall dependence of TxgGalNAc-T3^WT activity on Mn²⁺, with no activity observed in the presence of Ca²⁺ or Mg²⁺, similar to metazoan GalNAc-Ts (Fig. S7f)[35].

## An active site loop (II) modulates substrate binding

Metazoan GalNAc-Ts contain a catalytic flexible gating loop (loop I) that becomes ordered upon UDP-GalNAc binding and has an additional role in peptide binding[19,29,30]. Loop I is semi-conserved in TxgGalNAc-T3 (His414-Pro426, Figs. 4a, S4c). GalNAc-Ts contain an additional loop (Loop II) in the active site (Fig. 4b), and in TxgGalNAc-T3 and its apicomplexan homologs, it mainly consists of small hydrophobic residues (Ala317-Cys322) including Gly319, which makes a mainchain interaction with the peptide backbone (Fig. 4a). Loop II is disordered in the apo structure and becomes more ordered upon peptide substrate binding (Fig. 4a). Loop II flexibility and the interaction that occurs between the Gly319 backbone carbonyl and an amide group on both di-glycopeptide substrates (Muc5AC-3,13 and CST1.4) allows TxgGalNAc-T3 to accommodate and modify substrates with

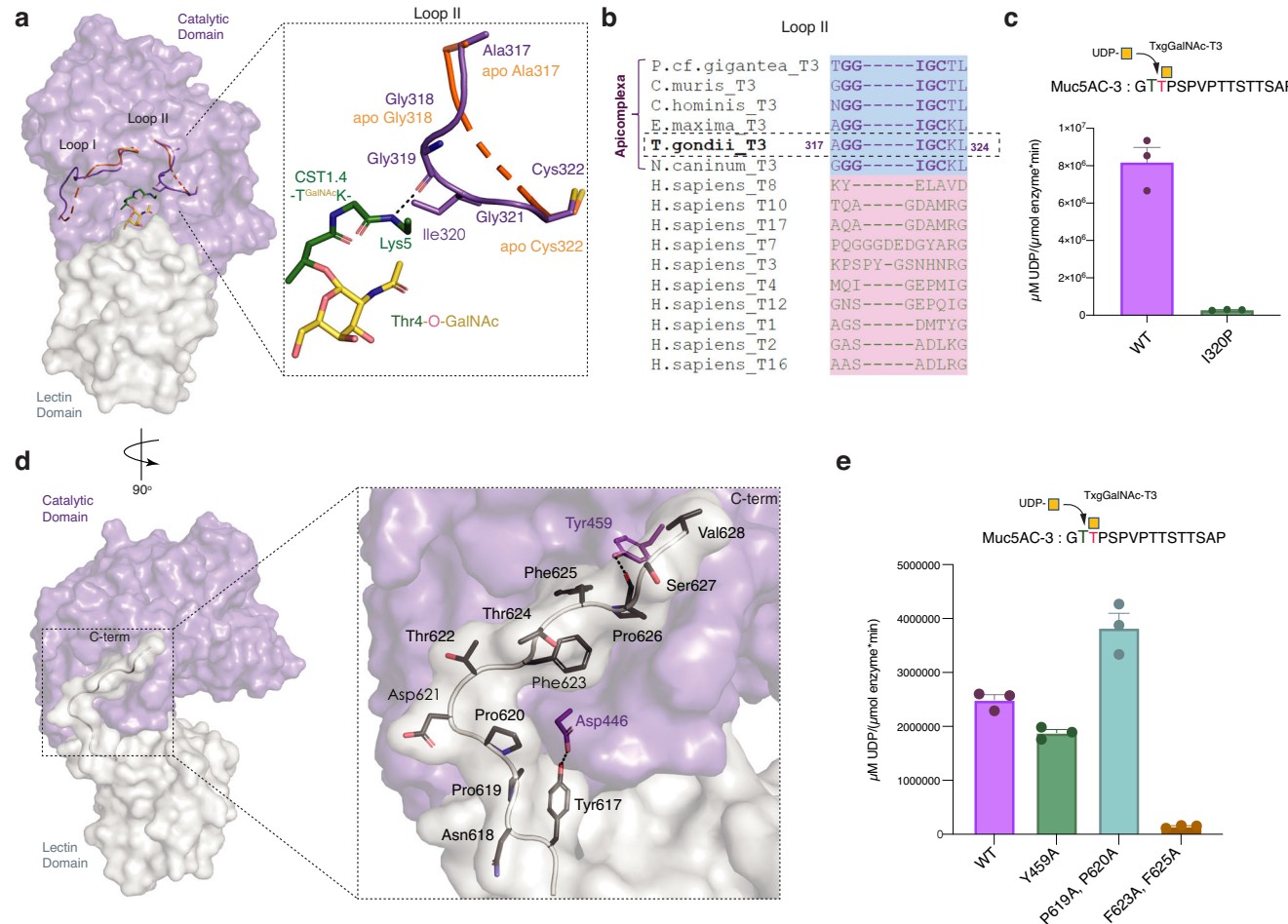

**Fig. 4 | Loop (II) and the C-terminal tail influence enzymatic function. a** GalNAc-Ts contain a catalytic gating loop that helps align UDP-GalNAc in the active site (Loop I, shown in purple for substrate bound TxgGalNAc-T3 and orange for the apo enzyme, see Fig. S4c for more details). In substrate bound TxgGalNAc-T3, loop II (purple) interacts with the peptide backbone. In the apo TxgGalNAc-T3, the loop is more disordered (orange). **b** Sequence alignment comparing loop II in TxgGalNAc-T3 to apicomplexan homologs and human GalNAc-Ts. Loop II is not conserved between apicomplexan (blue highlight) and human (pink highlight) isoenzymes and has not previously been associated with substrate binding. **c** Insertion of a Pro in the middle of loop II (I320P) disrupts enzymatic activity. **d** TxgGalNAc-T3 contains an extended C-terminal tail that makes hydrophobic and H-bonding interactions with the catalytic domain. **e** The P619A/P620A double mutation decreases loop rigidity and slightly increases enzymatic function. Reducing loop hydrophobicity (F623A/F625A) disrupts enzymatic function. For each variant in **c**, **e**, *n* = 3 biologically independent experiments performed in triplicate each time (9 technical replicates). The error bars in **c**, **e** represent the standard error of the mean (SEM). Source data are provided as a Source Data file.

variable sequences. Substituting the central loop residue Ile320 with Pro to perturb the conformation and stability of the loop diminishes enzymatic activity, supporting a possible role for loop II in substrate binding and alignment (Fig. 4c).

**A unique C-terminal tail influences TxgGalNAc-T3 function**

TxgGalNAc-T3 contains a non-conserved extended C-terminal tail (Tyr617-Val628) that interacts with the catalytic domain via hydrophobic and H-bonding interactions (Fig. 4d). Given the extensive nature of these interactions, we hypothesize that the extended C-terminus stabilizes the enzyme conformation and restricts the movement of the lectin domain. Tyr617 of the C-terminal tail H-bonds with Asp446 in the catalytic domain to hold the N-terminal end of loop in position, while residues Pro619-Pro620 behave like a rigid stand before the tail bends to position Phe623 and Phe625 into a hydrophobic cleft of the catalytic domain. Finally, an H-bond between Tyr459 in the catalytic domain with the mainchain of Pro626 holds down the C-terminal end of loop. Attempts to mutate Tyr617 and truncate the C-terminal tail abrogate protein expression in yeast, suggesting destabilization of TxgGalNAc-T3. TxgGalNAc-T3^F623A/F225A precipitates during purification and mostly elutes in the void volume on size exclusion

chromatography (SEC) (Fig. S8). TxgGalNAc-T3^Y459A has two melting events, the first once having a $T_M$ of 47.6 °C compared to 50 °C for the single melting temperature for TxgGalNAc-T3^WT (Fig. S8), while TxgGalNAc-T3^P619A/P620A behaves similarly to TxgGalNAc-T3^WT on SEC and Nano-DSF. We observed a ~40% reduction in activity for Y459A and significantly reduced activity for F623A/F625A (Fig. 4e). Interestingly, increasing the tail flexibility in the P619A/P620A variant results in a ~25% increase in activity compared to TxgGalNAc-T3^WT.

**The TxgGalNAc-T3 lectin domain does not influence activity**

The lectin domain consists of 3 repeats (α, β, and γ) each with the potential to interact with GalNAc if a conserved Asp-His-Asn GalNAc binding motif is present (Fig. S9a). In certain metazoan GalNAc-Ts, lectin domain interactions with a Thr-O-GalNAc ~9-12 amino acids away from the acceptor has long-range enhancing effects on activity[20]. Although none of the TxgGalNAc-T3 lectin repeats contain the Asp-His-Asn GalNAc binding motif, we wondered if TxgGalNAc-T3 uses a distinct mode of GalNAc recognition to enhance activity. However, we do not observe enhancement when comparing TxgGalNAc-T3 activity towards di-glycopeptides that contain one GalNAc at the +1 position and another GalNAc at either +12 (DGPI) or -10 (DGPII) when compared

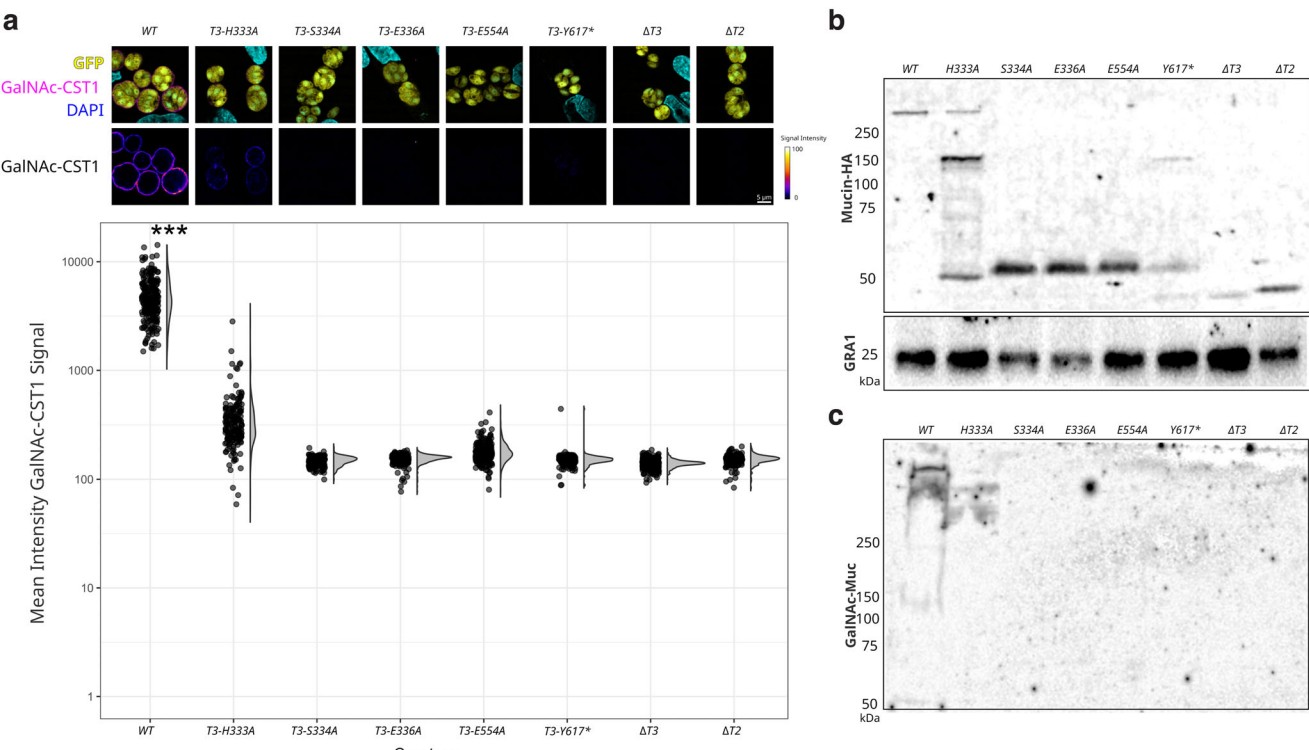

**Fig. 5 | The second metal site and C-terminal tail influence in vivo O-glycosylation. a** *T. gondii* cysts in HFF cells probed with GalNAc glyco-epitope-specific anti-CST1 mucin antibody reveal that the glycosylation of the cyst wall is diminished in TxgGalNAc-T3 mutant parasites. GFP expression indicates that the parasites are differentiated into cyst-forming bradyzoites. The lower panels delineate the reduced intensity of cyst wall glycosylation. Quantification of cyst wall glycosylation level by the GalNAc glyco-epitope-specific anti-CST1 mucin antibody. Analysis of variance across all genotypes was performed using a one-way ANOVA, yielding $p = 2 \times 10^{-16}$. *** Further analysis with a pairwise post-hoc analysis Tukey's HSD test between WT and all other genotypes yielded $p = 4.76 \times 10^{-11}$. The number of observations for genotypes are $n = 222, 182, 183, 263, 223, 173, 584$, and $190$ respectively. The total number of observations was $n = 2020$. **b** Immunoblot of TxgGalNAc-T3 mutant parasites expressing a surrogate mucin protein demonstrating the reduction of the degree of glycosylation indicated by a change in their molecular weight. The surrogate mucins were probed with an anti-HA antibody and protein loading was assessed with an anti-GRA1 antibody. Experiments were repeated three times with similar results. **c** A parallel immunoblot was probed with glyco-epitope-specific antibody to demonstrate the glycosylation of surrogate mucin proteins. High molecular bands bound to glyco-epitope antibody, but not intermediate or low molecular bands. Source data are provided in the Source Data file. The code for image quantification in **a** is provided as Supplementary Dataset 2.

to mono-glycopeptides with GalNAc only at the +1 position (GPIID). Thus, a role for the lectin domain in activity enhancement through interactions with a distal GalNAc on a di-glycopeptide has not been shown (Fig. S9B) but is consistent with the lack of electron density at the C-terminal Thr-O-GalNAc in the co-crystal structures of TxgGalNAc-T3 bound to the di-glycopeptides CST1.4 and Muc5AC-3,13 (Figs. 2a and S3a).

## Variations in TxgGalNAc-T3 influence in vivo glycosylation

To assess the significance of the in vitro characterizations of TxgGalNAc-T3 in an intact organism, *T. gondii* cell lines expressing various $Mn^{2+}_N$ and C-terminal tail mutants were generated by genetically manipulating the TxgGalNAc-T3 locus. Mutations in residues constituting the $Mn^{2+}_N$ site and a C-terminal tail truncation resulted in the disruption of cyst wall O-glycosylation as detected by a glycoepitope-specific antibody (Fig.5a). Interestingly, although TxgGalNAc-T3[H333A] results in diminished activity in vitro (Fig. 2d), it does not completely abolish cyst wall O-glycosylation in vivo, suggesting the presence of additional co-factors or conditions that influence TxgGalNAc-T3 function in vivo.

To evaluate the extent of glycosylation within the mucin domain of CST1, we introduced a surrogate protein comprising the CST1 signal peptide, CST1 mucin domain, and hemagglutinin (HA) epitope tag without SRS domains. Expression of protein in both TxgGalNAc-T3[WT] and various TxgGalNAc-T3 mutants and the resulting parasite lysates were analyzed by immunoblotting of the HA tag (Fig. 5b) and using a

glyco-epitope-specific antibody (Fig. 5c). In TxgGalNAc-T3[WT] cells, the mucin domain appears highly O-glycosylated, as evidenced by the migration pattern of the surrogate mucin, which remained at the interface between the stacking and resolving gel. In contrast, the H333A mutant exhibited intermediately glycosylated species with molecular weight of around ~200 kDa. Furthermore, both lower molecular weight species at ~55 kDa and glycosylated forms are detected, suggesting that TxgGalNAc-T3[H333A] retains partial enzymatic activity in vivo. As expected, the remaining mutants affecting the $Mn^{2+}_N$ binding site completely abolished TxgGalNAc-T3 enzymatic activity, as indicated by the presence of lower molecular weight species in the immunoblots. In the case of the C-terminal truncation mutant, only intermediately glycosylated species were generated, with highly glycosylated forms conspicuously absent. In summary, mutations in the second metal binding site and C-terminal region of TxgGalNAc-T3 disrupt bradyzoite cyst wall glycosylation, with varying effects on enzymatic activity and mucin domain glycosylation.

## Discussion

We report the first structures of a mucin-type O-glycosyltransferase from a protozoan pathogen, TxgGalNAc-T3, revealing unique features that are not conserved in metazoan homologs. The structures provide mechanistic insights into putative druggable sites to treat latent toxoplasmosis and other related parasitic diseases that are currently challenging to manage. In addition, we characterize the TxgGalNAc-T3 substrate preference for GalNAc at the +1 position of an acceptor on

substrate using a broad repertoire of (glyco)peptides and show that the transferase can fully glycosylate a stretch of Thr in vitro by a distributive mechanism. This preference is shared by human GalNAc-T10, T7, and T17, but these isoenzymes do not contain a similar motif and most likely use a distinct substrate binding pocket to interact with Thr-O-GalNAc. The structures reveal novel characteristics that are critical for the function of TxgGalNAc-T3 in vitro and in vivo and strictly conserved among its apicomplexan homologs, including a unique 2nd metal ($Mn^{2+}_N$), a novel active site residue (Glu332), a dynamic substrate binding loop (II), and an extended C-terminus (Fig. 6a, b).

$Mn^{2+}_N$ is coupled to the GalNAc binding pocket that dictates the TxgGalNAc-T3 + 1-substrate preference. $Mn^{2+}_N$ coordination by His333 and Glu554 does not align these sidechains for GalNAc binding, but instead is likely dictating GalNAc binding and catalysis by influencing charges on coordinating and adjacent residues. The second metal site is conceivably sensitive to metal concentrations in vivo, where changes in metal concentration during parasite invasion and replication could regulate glycosylation. While little is known about the role of $Mn^{2+}$ in *T. gondii*, $Ca^{2+}$ leakage into the cytosol from intracellular organelles such as the ER or Golgi, where GalNAc-Ts are anchored, or uptake of extracellular $Ca^{2+}$ modulates the lytic life cycle of tachyzoites and host invasion[36,37]. Thus, it is possible that the functions of a Golgi anchored TxgGalNAc-T is fine-tuned by $Ca^{2+}$ binding to the second metal site during host infection to upregulate or downregulate glycosylation, depending on the optimal and inhibitory concentrations of metal at the 2nd metal site. Additionally, while little is known about the function of TxgGalNAc-T3 in tachyzoites, both TxgGalNAc-T2 and -T3 are highly expressed during the virulent tachyzoite life cycle stage, leaving many unanswered questions regarding TxgGalNAc-T3 substrates, regulation, and role in the tachyzoite life cycle[24]. Finally, since $Ca^{2+}$ has less stringent metal coordination rules than $Mn^{2+}$, it may bind more efficiently in the absence of His333, which could explain why we see more glycosylation in vivo at physiological concentrations of $Ca^{2+}$ than we do in vitro. Other unknown factors could also contribute to the in vivo activity of TxgGalNAc-T3$^{H333A}$.

Our structures reveal an unexpected active site residue Glu332 that influences reaction chemistry. The current catalytic mechanism of human GalNAc-T2 is proposed to be $S_N$i type[38], where the acceptor Thr hydroxyl uses a front-face reaction to form a bond with the anomeric C1 carbon of the GalNAc moiety of the donor substrate, resulting in the retention of configuration. In the first step of catalysis, the hydroxyl hydrogen on the acceptor threonine and amide group hydrogen of GalNAc stabilize the negative charge that develops on the β phosphate oxygen of UDP as the bond between UDP and GalNAc breaks, forming an oxocarbenium ion (Fig. 6c). This is followed by a front face attack of the acceptor Thr at the C1 carbon of GalNAc to form Thr-O-αGalNAc. The reaction does not use a general base, and transfer of the hydroxyl hydrogen to the phosphate leaving group occurs as the bond between GalNAc and Thr forms.

In TxgGalNAc-T3, the acceptor Thr is similarly positioned for front face attack. However, the orientation and distance of Glu332 to the acceptor Thr positions it to act as a general base to deprotonate the acceptor Thr for a nucleophilic attack in an $S_N$2-type reaction, resulting in the inversion of configuration. This is puzzling since our Edman degradation sequencing shows that TxgGalNAc-T3 produces Thr-O-αGalNAc. The UDP-GalNAc donor would have to be a β anomer for inversion of configuration to yield Thr-O-αGalNAc, and there are currently no known enzymes that synthesize a β anomer of UDP-GlcNAc/GalNAc.

A more plausible mechanism is a double displacement mechanism (Fig. 6d), which was recently proposed for two distinct 3-deoxy-D-manno-oct-2-ulosonic acid (β-Kdo) glycosyltransferases[39,40]. In this case, Glu332 acts as a nucleophile and initiates catalysis by first forming a bond with the anomeric carbon (C1), resulting in inversion of configuration. This would involve a >2.5 Å approach towards C1, since

Glu332 is currently ~5.5 Å away from C1 on GalNAc (Fig. 2e). The acceptor Thr then initiates a second nucleophilic attack at C1, resulting in the retention of configuration in the final product (Thr-O-αGalNAc), which is consistent with our experimental results. The double-displacement mechanism is supported by our model of how $Mn^{2+}_N$ influences catalysis (Fig. 3d–f) since Glu332 is on a loop adjacent to His333, whose interactions with $Mn^{2+}_N$ and Thr-O-GalNAc could influence loop rigidity and conformation. At high pH (9), metal binding is tight, which would restrict the loop and prevent Glu332 from approaching C1. At pH 7, metal binding is less tight due to the protonation of His333, possibly making the loop more dynamic and allowing Glu332 to approach C1 for a nucleophilic attack. The proximal $Mn^{2+}_N$ could have an electron-withdrawing effect on Glu332 and help lower its pKa to make it a better nucleophile. Our ongoing QM and QM/MM simulations, along with additional structural and biochemical experiments, will provide further insights into the catalytic mechanism of TxgGalNAc-T3.

The other differences between TxgGalNAc-T3 and its metazoan homologs include the ordering of loop II due to non-specific backbone interactions with substrates, helping to align various substrates across the active site, while the C-terminal tail stabilizes an active conformation of the enzyme. Loop II is more variable in metazoan GalNAc-Ts, and in most crystal structures is not contacting the substrate (Fig. 4b). The exceptions are human GalNAc-T2 in complex with an unglycosylated peptide EA2, where loop II makes mainchain interactions with the peptide backbone, and GalNAc-T12, where the sidechain of Asn270 in loop II forms part of the GalNAc binding pocket and enhances glycopeptide substrate binding[19,29]. Whether these residues are required for substrate binding in human GalNAc-T2 is not clear, but the loop is ordered and not contacting the substrate in complex structures of GalNAc-T2 with glycopeptides, suggesting it may enhance peptide, but not glycopeptide binding.

Intriguingly, the lectin domain does not enhance activity by binding to distant sites, as seen with metazoan enzymes, but instead contains Glu554, which forms part of the +1 Thr-O-GalNAc binding pocket and indirectly interacts with $Mn^{2+}_N$. In addition, the C-terminal tail is part of the lectin domain, which helps to position it in the catalytic domain hydrophobic cleft. The possible inability of the TxgGalNAc-T3 lectin domain to enhance activity by binding a distal GalNAc is consistent with the sequential mechanism it uses to efficiently glycosylate stretches of Thr without needing to bind a long range GalNAc. Whether the lectin domain has additional functions in TxgGalNAc-T3 such as binding substrates or co-factors remains to be studied.

There are currently no treatments that can eliminate *T. gondii* tissue cysts and new medications to eradicate latent toxoplasmosis are clearly needed. Inhibition of human GalNAc-Ts has been of great interest given their roles in diseases such as cancer, but marked by many challenges, including finding selective inhibitors and using appropriate screening systems as described for GalNAc-T11[41]. Nevertheless, an appropriate glycoengineered mammalian cell line was successfully utilized for identifying inhibitors of human GalNAc-T3[42]. Other challenges in finding inhibitors include the lack of structural information for most human GalNAc-Ts, including GalNAc-T11. In contrast to mammalian cell lines, *T. gondii* has a simpler O-glycosylation system with fewer isoenzymes, and the structures and supporting data in this paper could facilitate the use of structure-based design of inhibitors.

The differences between TxgGalNAc-T3 and human GalNAc-Ts represent regions that can be potentially targeted for therapeutic purposes. Targeting TxgGalNAc-T3 could possibly weaken cyst walls and allow drugs that target bradyzoites to permeate the highly glycosylated barrier or permit the immune system to clear the cysts and latent infection to prevent reactivation to tachyzoites, an area we are currently exploring (Fig. 6a, b). Moreover, targeting TxgGalNAc-T3

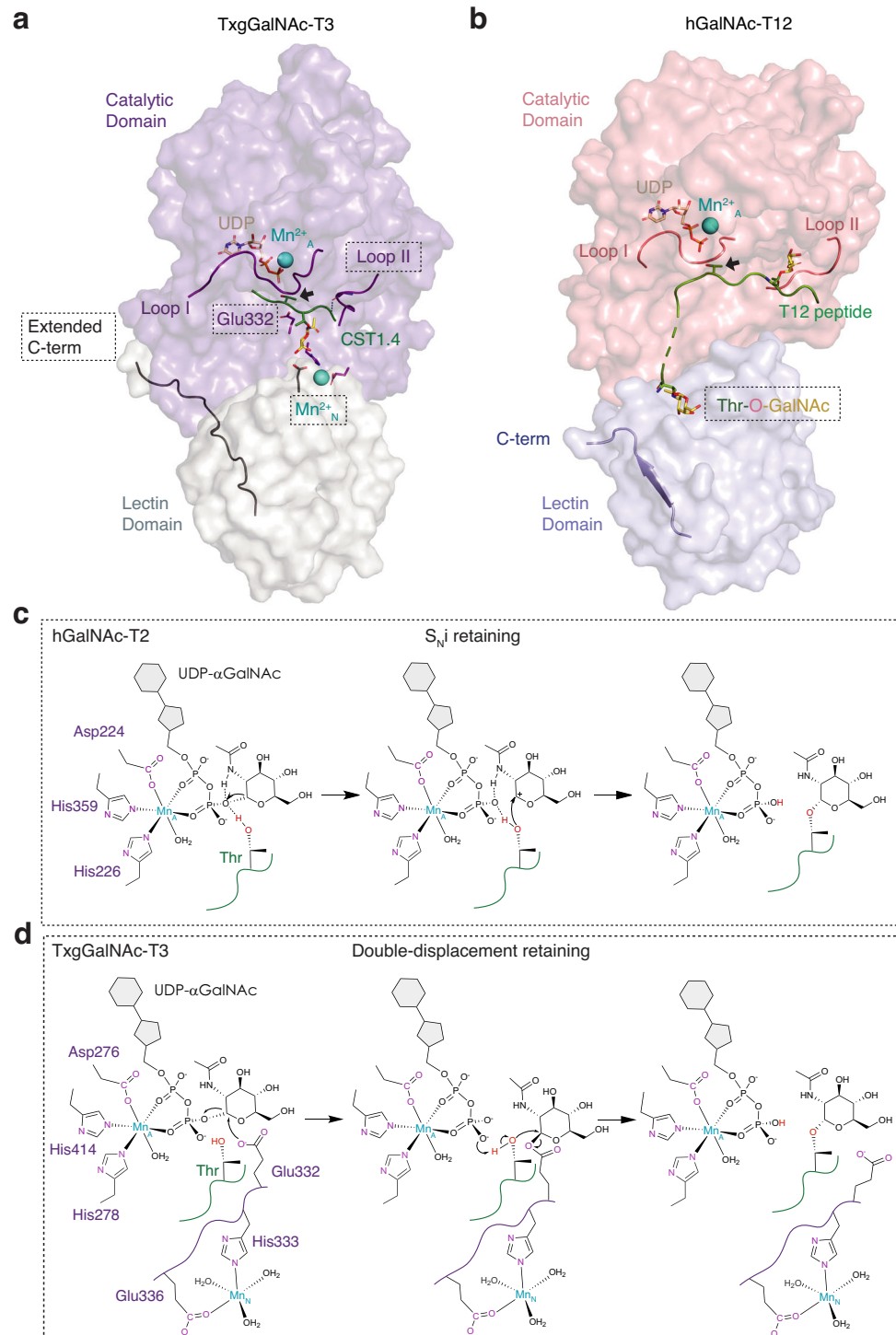

**Fig. 6 | Proposed catalytic mechanism of TxgGalNAc-T3. a** Structure of TxgGalNAc-T3 (catalytic domain in lavender and lectin domain in light grey) reveals unique characteristics, including a second $Mn^{2+}_N$ (aquamarine), a charged residue Glu332 in the active site close to the acceptor Thr, a flexible loop II (purple) that interacts with the substrate peptide mainchain, and an extended C-terminal tail (black). **b** Human GalNAc-T12 (catalytic domain in salmon and lectin domain in light blue) is missing these features but contains a GalNAc binding pocket in the lectin domain, which has not been shown to be present in the lectin domain of TxgGalNAc-T3. **c** $S_N i$ retaining catalytic mechanism of a human GalNAc-T (hGalNAc-T2). Here, the acceptor Thr approaches the anomeric carbon in a front-face reaction, resulting in the formation of an oxocarbenium ion. The β phosphate on the UDP leaving group extracts a proton from Thr as a bond forms between the acceptor and GalNAc with retention of configuration. **d** Proposed double-displacement catalytic mechanism of TxgGalNAc-T3. Glu332 could act as a nucleophile and initiate catalysis by approaching the C1 carbon on GalNAc for a nucleophilic attack, resulting in inversion of configuration to a β−linked GalNAc. The acceptor Thr then forms a bond with the anomeric carbon of GalNAc in an $S_N 2$ type nucleophilic reaction to displace Glu332 and retain the α stereochemistry on GalNAc in the final product.

could have direct detrimental effects on tachyzoites, although this remains an open question. Given the similarities between TxgGalNAc-T3 and its apicomplexan homologs, it is conceivable that inhibitors of TxgGalNAc-T3 could be used more widely to target other disease-causing parasites, such as *Cryptosporidium hominis* which causes the gastrointestinal infection cryptosporidiosis, as well as *Neospora caninum* and *Eimeria maxima*, which infect livestock resulting in agricultural economic losses. In conclusion, our studies of TxgGalNAc-T3 have shed light on the evolution of this enzyme family and lay the groundwork for future studies on anti-microbials that target toxoplasmosis and other parasitic diseases.

## Methods

### Expression and purification of TxgGalNAc-T3

TxgGalNAc-T3 (aa 74–635) was cloned from a TxgGalNAc-T3 cDNA template (GenScript AY160970.1, GeneID:7901231, TGME49_318730) into the expression vector pPICZα-A (Invitrogen) for secreted protein expression in *Pichia pastoris*. TxgGalNAc-T3 mutant constructs were made by site-directed mutagenesis (Table S2, all cloning primers are included in Supplementary Dataset 3). For strain construction, cloned plasmids were linearized with the restriction enzyme PmeI and transformed into *Pichia pastoris* SMD1168 cells (Invitrogen) by electroporation. To express TxgGalNAc-T3, cells were grown at 30 °C to an $OD_{600}$ ~ 20 in Buffered Glycerol Complex Media (BMGY), containing (2 % (w/v) peptone, 1% (w/v) yeast extract, 1.34% (w/v) Yeast Nitrogen Base (YNB), 4 ×10$^{-5}$ % (w/v) biotin, 1% (v/v) glycerol, 100 mM potassium phosphate pH 6.0) and 100 µg/ml of Zeocin (Invivogen). To induce protein expression, cells were cultured by centrifugation (1500 X g for 10 min) and resuspended in 1/5 volume Buffered Methanol Complex Medium media (BMMY, where 1% glycerol is replaced with 0.5% methanol) followed by incubation/shaking at 20 °C for 24 h.

Cells were centrifuged (1500 X g for 10 min) and supernatant was collected, filtered, and pH adjusted by adding 50 mM Tris pH 8.0, 250 mM NaCl, 5% glycerol and 10 mM β-mercaptoethanol (βME). Purification was carried out at 4 °C. Supernatant was applied onto a 5 ml HisTrap HP column (Cytiva) pre-equilibrated with 5 column volumes (CV) of buffer A (25 mM Tris, 250 mM NaCl, 10 mM βME and 5% glycerol, pH 7.5). Protein was eluted by a 50-500 mM imidazole gradient over 10 CV. Peak fractions were pooled and incubated with TEV protease at a ratio of 1:20 (w/w) while dialyzing into 500 ml of buffer A containing 25 mM imidazole at 4 °C. The His-tagged TEV protease and residual uncut protein were removed by manually loading the sample onto a 1 ml HisTrap HP column (Cytiva) equilibrated with 5 CV dialysis buffer, followed by washing with 1 CV of dialysis buffer. After pooling untagged TxgGalNAc-T3 in the flow-through and wash samples and increasing glycerol concentration to 30%, protein was aliquoted, flash frozen in LN2, and stored at −80 °C.

### UDP-Glo enzymatic assay

TxgGalNAc-T3 (WT and variants) were thawed and exchanged into assay buffer (25 mM HEPES, 100 mM NaCl, 0.5 mM EDTA, 5% glycerol and 10 mM βME, pH 7.3) by centrifugation in a 10 kDa cut-off Amicon ultra-concentrator at 4000 x g (Millipore Sigma). Protein concentration was estimated by Pierce™ BCA Protein Assay kit (ThermoFisher Scientific). Glycosyltransferase activity was assayed using a UDP-Glo™ Glycosyltransferase Assay kit (Promega, V6961). A 25 µL reaction was initiated by adding 50 nM and 100 nM of purified WT and variant TxgGalNAc-T3, respectively, to a 5 mM donor substrate uridine diphosphate (UDP)-N-acetylgalactosamine (GalNAc) (supplied with the kit) and the following acceptor substrates: Muc5AC, Muc5AC-3, Muc5AC-13, Muc5AC-3,13, CST1.1, CST1.2, CST1.3, CST1.4, SRS13.1, SRS13.2, SRS13.3, and SRS13.4 peptides and glycopeptides (AnaSpec, Fremont, CA) in 5 mM $MnCl_2$ (Sigma Aldrich) and 1 X HEPES buffer (100 mM NaCl, 5 mM βME, 25 mM HEPES, pH 7.3).

Reactions were assembled in a 96-well plate (Corning) and incubated at 37 °C for 15 min. The reaction was terminated by adding 25 µL UDP detection reagent to each well followed by incubation at 27 °C for 60 min and luminescence was recorded using a Synergy Neo2 (Biotek) plate reader. A standard curve of UDP in 5 mM $MnCl_2$ and 1X HEPES buffer was applied at each measurement to associate a defined UDP concentration with a luminescence signal that directly correlates with the glycosyltransferase activity within 60 min under described conditions. Multiple reactions with varying substrate concentrations (0-2000 µM) were used to determine the kinetic parameters of the glycosyltransferase reactions. All reactions were performed in triplicate 3 independent times ($n = 3$, Source Data file). Data were analyzed with Microsoft Excel and kinetic parameters were calculated using GraphPad Prism (Boston, MA). Remarkably, TxgGalNAc-T3 shows no UDP-GalNAc hydrolase activity against these glycopeptides as demonstrated by the Sephadex G10 chromatography (Fig. S10) performed for the Edman sequencing experiments in Fig. 1d and supplemental Fig S2. However, significant hydrolase activity is observed in the Sephadex G10 chromatography (Fig. S10) of random glycopeptide substrates lacking or partially lacking acceptor sites N-terminal of a Thr-O-GalNAc, i.e. GPIIC Fig. 1b and glycopeptides in supplemental Fig. S9b.

### Nano-DSF

TxgGalNAc-T3 variants were thawed and exchanged into assay buffer (25 mM HEPES, 100 mM NaCl, 0.5 mM EDTA, 5% glycerol and 10 mM βME, pH 7.3) on a Superdex 200 Increase 10/300 GL (Cytiva) followed by concentration using a 10 kDa cut-off Amicon ultra-concentrator at 4000 x g (Millipore Sigma). Concentration was estimated by Pierce™ BCA Protein Assay kit (ThermoFisher Scientific). Denaturation profiles of the wild type (0.4 mg/ml) and variants H333A (0.19 mg/ml), H333N (0.4 mg/ml), S334A (0.26 mg/ml), E336A (0.4 mg/ml), E336D (0.4 mg/ml), E336Q (0.095 mg/ml), E554A (0.231 mg/ml), Y459A (0.4 mg/ml), P619A/P620A (0.4 mg/ml), and F623A/F625A (0.086 mg/ml) were obtained by using Tycho NT.6 (NanoTemper GmbH, Germany). Protein samples (~ 10 µl) were loaded into NT.6 glass capillaries and heated from 35 °C to 95 °C at a rate of 30 °C/min. Raw datasets contain fluorescence intensity at 330 nm (F330) and 350 nm (F350), their ratio (F350/F330), and their first derivatives ($\partial F330/\partial T$, $\partial F350/\partial T$, $\partial(F350/F330)/\partial T$). First derivatives (F350/F330) vs. Temperature (°C) graphs were plotted in GraphPad Prism.

### Probing neighboring glycosylation activity against glycopeptide GPIIC

Assays against random glycopeptides GPIIC (Fig. 1b) (Sussex Research, Ottawa, CN) consisting of 100 µl reactions in 100 mM sodium cacodylic pH 6.5, 0.8 mM β-mercaptoethanol, 0.08% Triton X100, 10 mM Mn$^{2+}$, ~ 0.8 mM GPIIC glycopeptide, 0.4 mM UDP-[$^3$H]GalNAc, 1.4 µM TxgGalNAc-T3, were incubated at 37 °C on a shaking microincubator for 5 hr and 21 hr. Reactions were quenched with 100 µl of 250 mM EDTA, diluted to 4 ml and passed over a 1 ml column of DOWEX 1×8 (Sigma Aldrich). Dowex column flow through was lyophilized and applied to a Sephadex G10 column (Cytiva) to separate peptide from free [$^3$H]GalNAc and the peptide peak was lyophilized for sequencing. Glycopeptide products were Edman sequenced on a modified gradient Shimadzu PPSQ53A sequencer (Shimadzu Scientific instruments Inc., Columbia, MD). The $^3$H-GalNAc-O-Ser-PTH derivatives (8.5-12 min) at each cycle were collected on a Shimadzu FRC-10A fraction collector and $^3$H-scintillation counted on a Beckman LS6500 scintillation counter.

### Probing lectin domain long range glycosylation activity against di-glycopeptides

To probe lectin domain interactions, 100 µl reactions consisting of 100 mM sodium cacodylic pH 6.5, 0.8 mM β-mercaptoethanol, 0.08 % Triton X100, 10 mM Mn$^{2+}$, ~ 1.7 mM random glycopeptides DGPI, DGPII

and GPIID (Sussex Research, Ottawa, CN), 0.1 mM UDP-[³H]GalNAc, 0.7 μM TxgGalNAc-T3, incubated at 37 °C for 5 hr. Reactions were quenched with 100 μl 250 mM EDTA and processed as described for GPIIC. Pre- and post- Dowex ³H DPM and ³H DPM integration of the Sephadex G10 glycopeptide and GalNAc peaks were used to calculate percent of glycopeptide glycosylated and percent of UDP-[³H]GalNAc hydrolysis.

## Identification of sequential glycosylation sites in *T. gondii* peptides and Muc5AC by Edman sequencing

For O-glycosylation analysis of glycopeptides CST1.2, CST1.3, CST1.4, SRS13.2, SRS13.4, Muc5AC-3, Muc5AC-3,13 and Muc5AC-13, (AnaSpec, Fremont, CA) by Edman Sequencing, 41-46 μl reactions consisting of 110-120 mM sodium cacodylic pH 6.5, 0.8 mM β-mercaptoethanol, 0.08% Triton X100, 12 mM $Mn^{2+}$, 0.5-0.6 mM (glyco)peptides, 2.2-2.4 mM UDP-[³H]GalNAc, and 0.06-0.15 μM TxgGalNAc-T3 were incubated at 37 °C for 30 min (CST1.x) or 90 min (SRS13.x & Muc5ACx). Overnight incubations were also performed for CST1.3 and Muc5AC-13. Reactions were quenched with 100 μl 250 mM EDTA and processed as described for GPIIC. Pre and post Dowex ³H DPM and ³H DPM integration of the Sephadex G10 glycopeptide and GalNAc peaks were used to calculate percent of glycopeptide glycosylation and revealed no UDP-[³H]GalNAc hydrolysis against these glycopeptides. Glycopeptide products were Edman sequenced on a modified gradient Shimadzu PPSQ53A sequencer. Free GalNAc (2.25-3.25 min) and the GalNAc-O-Thr-PTH and GalNAc-O-Ser-PTH derivatives (9.0-14.5 min) at each cycle were collected on a Shimadzu FRC-10A fraction collector and each fraction scintillation counted on a Beckman LS6500 scintillation counter. Sequence chromatograms of the PTH (phenylthiohydantoin) derivatives were also analyzed as described in Supplemental Fig. S2j, k.

## Kinetics of TxgGalNAc-T3 against Ser-O-GalNAc and Thr-O-GalNAc glycopeptides

Stock solutions of glycopeptide substrates T7T* and T7S* (Sussex Research, Ottawa, CN) were made to yield final reaction concentrations of 1.4 mM, 0.7 mM, 0.35 mM, 0.175 mM, and 0.088 mM. Reactions consisted of 100 mM sodium cacodylic pH 6.5, 1 mM β-mercaptoethanol, 0.1% Triton X-100, 2 mM UDP-[³H]GalNAc, 0.044 μM enzyme, peptide substrate, and were incubated at 37 °C. Reaction times varied, depending on substrate concentrations, ranging from 10 to 30 min to maintain peptide glycosylation to <20%. After incubation, reactions were quenched with 200 μL of 0.5% TFA in $H_2O$. BioPureSPIN TARGA-C18 spin columns (The Nest Group, Ipswitch MA) were pre-equilibrated by passing sequentially: acetonitrile (300 μL), 50/50 acetonitrile/$H_2O$ in 0.1% TFA (300 μL), and 0.1% TFA in $H_2O$ (700 μL). The latter washes were eluted by spinning at 800 rpm in an Eppendorf Minispin Plus tabletop centrifuge. Ten percent (22 μL) of the reaction volume was removed for [³H] scintillation counting (initial DPM), and the remainder was applied to the equilibrated TARGA C18 hydrophobic spin columns and spun for 1 minute.

After the sample was eluted columns were washed with 800 μL of 0.1% TFA in $H_2O$ to remove free UDP-[³H]GalNAc and [³H]GalNAc by centrifugation at 36 x *g* (800 rpm) for 1 min giving the A eluate. The bound (glyco)peptide products/reactants (B eluate) were eluted using two washes of 200 μL of 50/50 acetonitrile in 0.1% TFA followed by 200 μL of 100% acetonitrile, each spun for 1 min at 36 x *g* (800 rpm), and a final 100 μL of 100% acetonitrile spun for 4 min. [³H] scintillation counting was performed on the combined flow through and wash (A eluate) and the eluted (glyco)peptide products/reactants (B eluate). The extent of glycosylated peptide, in mM, was obtained by dividing the B counts (in DPM) of the eluted (glyco)peptides by the initial total DPM (as well as by the sum of the DPM of the A and B eluates) of the UDP-[³H]GalNAc and by multiplying by the initial mM of UDP-GalNAc. Values were converted to μM of GalNAc transferred/(μM enzyme*min) according to the initial amount of UDP-GalNAc, substrate, and enzyme

used. $k_{cat}$ (μM GalNAc/(μM enzyme*min) or min⁻¹), $K_M$ (μM), and $V_{max}$ (μM GalNAc/min) values and Michaelis Menten plots were obtained using GraphPad Prism software (Boston, MA).

## Crystallization, data collection and processing, structure determination and refinement

TxgGalNAc-T3 was thawed and exchanged into crystallization buffer (25 mM HEPES, 100 mM NaCl, 0.5 mM EDTA, 5% glycerol and 10 mM βME, pH 7.3) on a Superdex 16/600 HiLoad column (Cytiva). Peak fractions were concentrated using a 10 kDa cut-off Amicon ultra-concentrator (Millipore Sigma) at 4000 x g to ~10-15 mg/ml. Each enzyme-peptide-UDP-$Mn^{2+}$ complex was prepared by combining TxgGalNAc-T3, 5 mM of one of the glycopeptides (Muc5AC-3, Muc5AC-13, Muc5AC-3,13, CST1.4, and SRS13.2 (AnaSpec, Fremont, CA), 5 mM UDP-2-(acetylamino)-4-F-D-galactosamine disodium salt UDP-GalNAc-F, (Chembind, Atlanta, GA), and 5 mM $MnCl_2$ to a final protein concentration of 6.0 mg/ml. Hanging drops were prepared by mixing 1 μl of protein complex solution with 1 μl of reservoir solution containing 0.1 M CHES pH 9.5 and 14-20% PEG 8000 (w/v) and equilibrated against a 500 μl reservoir solution. Crystals formed after 4 days in 24-well plates incubated at 20 °C. Crystals were cryoprotected in a crystallization solution containing 20% glycerol and flash frozen in LN2 prior to X-ray data collection.

X-ray data was collected at the Advanced Photon Source SER-CAT ID and BM-22 beam lines (Argonne, IL). HKL2000 was used to process and scale the X-ray diffraction data (Tables S1 and S4)[43]. The initial structure was solved by molecular replacement using Phaser (CCP4) and an Alphafold2 model of TxgGalNAc-T3 as an initial search model[44–46]. Initial models were rebuilt manually using Coot and refined in PHENIX[47,48]. The final models were validated by using PROCHECK and MOLPROBITY[49–52]. Structure figures were prepared with Pymol (The PyMOL Molecular Graphics System, Version 2.0 Schrodinger, LLC) and ChimeraX. Structural and sequence analyses were performed with Clustal Omega (through the ChimeraX GUI) and ChimeraX[53].

## Quantum chemistry

From the crystal structure of TxgGalNAc-T3 in complex with a glycopeptide substrate, we chose His333, Ser334, Tyr335, Glu336, Glu554 and a portion of glycopeptide substrate as well as the $Mn^{2+}$ with a spin multiplicity of 6 and water molecules for quantum chemical calculations at the level of density functional theory. Their backbone and side chains were then modified; see the coordinates given in the Supporting files. Quantum chemical calculations were carried out with Gaussian 16[54] on the neutral and protonated His333. We employed M06-L with the basis set of cc-PVDZ in the water reaction field for geometry optimization.

## *T. gondii* cell culture and strains

Prugniaud strain with a deletion in KU80 gene[55] was cultured in human foreskin fibroblasts (HFF) in 10% fetal bovine serum (FBS) in Dulbecco's modified Eagle medium (DMEM) with penicillin-streptomycin at 5% $CO_2$. For the induction of bradyzoite differentiation, DMEM with 1% FBS with 25 mM HEPES adjusted to pH 8.2 was used at atmospheric $CO_2$[7].

## Genetic manipulation of *T. gondii*

For generating point mutations in the TxgGalNAc-T3 gene in *T. gondii*, gRNAs targeting the TxgGalNAc-T3 locus was used with the donor oligos that repair with the desired point mutation[8,56]. Two candidate gRNA sequences were selected for three loci (His333/Ser334/Glu336, Glu554, and Tyr617). The 70-base single stranded gRNA oligonucleotides were designed by the selected gRNA sequences flanked by 25-base homology sequences to the gRNA-Cas9 vector. The gRNA oligonucleotides were incorporated into a linearized gRNA-Cas9 vector using NEBuilder HiFi DNA Assembly kit (New England Biolabs).

Construction of the gRNA-Cas9 vectors were verified with Sanger sequencing of the gRNA locus. Each donor oligo contains a point mutation that replaces original codon with alanine or stop codon flanked by 40 nt homologous recombination sequences on both 5' and 3' (all donor sequences and gRNA-Cas9 vectors are shown in supplementary Dataset 3). The gRNA-Cas9 vectors and corresponding donor oligos were electroporated into the parental Pru strains and subcloned by limiting dilution. The point mutations were verified by Sanger sequencing the genomic DNA of the parasite clones. Surrogate mucin construct was generated by NEBuilder by concatenating the constitutive promoter, CST1 signal peptide, CST1 mucin domain, 3x HA sequences, and selectable marker DHFR. The surrogate mucin construct was electroporated into the parasite and integrated into genome by pyrimethamine selection.

## Immunofluorescence assay and immunoblotting

HFF cells, grown on a coverslip infected with TxgGalNAc-T3 point mutant *T. gondii*, were cultured in a bradyzoite differentiation medium for 72 h[7]. Following incubation, cells were fixed with 4% paraformaldehyde in PBS for 30 minutes and subsequently permeabilized with 0.2% Triton-X100 in PBS for 20 minutes. For immunostaining, cells were incubated with a 1:200 dilution of rabbit anti-GFP antibody (ThermoFisher #G10362) and 1:200 dilution of a GalNAc glycoepitope-specific anti-CST1 antibody[7], both prepared in PBS with 1% BSA. The incubation was carried out for 90 minutes at 37 °C. Secondary fluorescent antibodies were applied at a 1:2000 dilution in PBS with 1% BSA for another 90 minutes. Quantification of glycosylation was conducted using a rabbit polyclonal CST1 antibody (1:200 dilution) raised against its SRS domain[7] to identify all CST1 and the glycoepitope-specific anti-CST1 antibody to measure the glycosylation level. Mean fluorescence intensity was calculated by the intensity of glycoepitope specific CST1 signal in CST1 positive area detected by polyclonal CST1 antibody. About 200 images were taken per genotype using a Leica SP8 confocal microscope. A custom Jython with ImageJ and R script was used for quantification and statistical analysis (included as Supplementary Dataset 2).

For immunoblot analyzes, HFF cells infected with *T. gondii* strains expressing the surrogate mucin protein with an HA tag were cultured for 48 hours in standard medium (10% FBS in DMEM). Cells were harvested, lysed in Laemmli SDS buffer, and the lysates subjected to SDS-PAGE. Following transfer, the blots were probed with HRP-conjugated rat anti-HA antibody (clone 3F10, Roche) at a 1:1000 dilution and incubated overnight. For a loading control and anti-GRA1 antibody was used at a 1:500 dilution. Signal detection was accomplished using the SuperSignal West Pico Plus reagent (ThermoFisher) and a LiCOR imaging system (LiCOR Bioscience).

## Reporting summary

Further information on research design is available in the Nature Portfolio Reporting Summary linked to this article.

# Data availability

All data generated or analysed during this study are included in this published article (and its supplementary information files) or are available from the corresponding author upon request. Structure coordinates and X-ray diffraction data have been deposited in the Protein Data Bank, www.wwpdb.org (PDB ID codes: 8UJG, 8UJH, 8UJF, 8UJE, 8UI6, 8UHV, 8UHZ, 8UI1). Source data are provided with this paper.

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

## Acknowledgements

We thank the beamline staff at the Advanced Photon Source for assistance. The quantum chemical study utilized the computational resources of the NIH HPC Biowulf cluster (http://hpc.nih.gov). Structural analyses were performed with UCSF ChimeraX, developed by the Resource for Biocomputing, Visualization, and Informatics at the University of California, San Francisco, with support from National Institutes of Health R01-GM129325 and the Office of Cyber Infrastructure and Computational Biology, National Institute of Allergy and Infectious Diseases. This work was supported by National Institutes of Health Grant 1-ZIA-DE000754-03 (to N.L.S.), 5-R01-GM113534-08 (to T.A.G.), and R01-AI134753 (to L.M.W.).

## Author contributions

P.K. and N.L.S. conceptualized the project. P.K. cloned, expressed, and purified TxgGalNAc-T3. P.K. crystallized TxgGalNAc-T3 and co-complexes and conducted in vitro biochemical assays. N.L.S. and P.K. harvested crystals and collected data, and P.K. processed the data and solved and refined the structures. T.A.G. conducted $^3$H-GalNAc assays on glycopeptide substrates and glycosylation site analysis by Edman sequencing and C.J.B. performed kinetic studies on the Ser and Thr glycopeptides. T.T. and L.M.W. performed the in vivo O-glycosylation

studies. Y.S.L. carried out quantum chemical calculations. N.L.S. prepared the manuscript. All authors contributed to the results and methods section and in editing the manuscript.

## Funding

## Competing interests

The authors declare no competing interests.
