## [Peer Review File · Nature Communications]

REVIEWER COMMENTS

Reviewer #1 (Remarks to the Author):

This study describes the substrate specificity and structure of a *Toxoplasma gondii* GalNAc-transferase-3 that is required for O-glycosylation of proteins in the bradyzoite cyst wall. Enzymes involved in the assembly of the heavily glycosylated cyst cell wall are considered potential drug targets for latent parasite stages. Initial studies confirmed that recombinant TgGalNAc-T3 transfers GalNAc to Thr residues in singly/multiply glycosylated peptides. GalNAc-T3 crystal structures in complex with an analogue of UDP-GalNAc and substrate glycopeptides revealed that the TgGalNAc-T enzyme has the same GT-A fold and conserved active site residues as mammalian GalNAcTs. Some differences were observed between the *Toxoplasma*/Apicomplexan GalNAc-Ts and mammalian GalNAc-Ts, including the presence of a second Mn²⁺ binding site (which may help binding to GalNAc), a surface exposed Loop II (which may modulate substrate binding) and a C-terminal tail (which binds to the catalytic domain). The functional importance of these domains in catalysis was supported by targeted mutagenesis studies. Finally, introduction of some of these mutations into the endogenous *T. gondii* GalNAc-T gene resulted in reduced cell wall O-glycosylation of in vitro-differentiated bradyzoite cysts, as well as reduced O-glycosylation of a secreted reporter protein, although no affect on bradyzoite differentiation was observed. This is a very nicely written and technically proficient study which represents a significant advance in our understanding of the structure and catalytic mechanism of this family of glycosyltransferases. The novelty of the findings are somewhat reduced by the previous studies which have already highlighted the importance of these enzymes for *T. gondii* cyst formation, and the lack of evidence that GalNAc-T3 is indeed druggable either in vitro or in vivo. In particular, the lack of a strong phenotype on bradyzoite differentiation following mutagenesis of key GalNAc-T3 residues and loss of cyst O-glycosylation suggests that any inhibitor will have to result in complete inhibition - a high bar that may not be achievable.

Major comments

Is metal binding site specific for Mn²⁺, or can it be substituted with Mg²⁺ or Ca²⁺? Have the authors measured enzyme activity following removal of all cations and sequential substitution with different cations?

Fig 7. It is difficult to assess the extent to which mutations in GalNAc-T3 lead to reduced cyst wall glycosylation based on the images provided, as staining with the GalNAc-CST Ab is already weak in WT (bottom 20% of signal sensitivity). Can the authors quantitate GalNAc-CST1 fluorescence in the different bradyzoite preparations?

Fig S7 is an important result and should be in the main text. However, the authors should also provide some indication of expression levels of the different mutated forms of GalNAc-T3. For example, do the mutations lead to loss of activity due to protein misfolding and clearance or due to reduced enzyme activity of equally expressing proteins.

How was bradyzoite differentiation performed (currently no reference)? The standard bradyzoite differentiation protocol utilizes high pH, which the authors have shown also results in loss of GalNAc-T3 activity. Is it possible that mutated proteins are just less stable than the WT protein at alkaline pH

and inactivated by the harsh differentiation conditions?

Discussion -line 351– the authors suggest that TgGalNAc-T3 can act in a processive mechanism to successively add GalNAc residues to a peptide. It is not clear what they mean by processive. When applied to GTs, it normally means that the substrate remains bound to the enzyme while different sugars are added. However, this is contrary to what the authors proposed based on their substrate-specificity trials (Line 159).

There are no inhibitors for GalNAc-T's at the moment despite significant interest and a recent study has suggested that it may be difficult to identify inhibitors of GalNAc-T1 (Nature Comms PMID: 36804936). This should perhaps be discussed.

Reviewer #2 (Remarks to the Author):

This is a very good paper with a number of interesting structures. Overall the paper contains a lot of novel information and the biochemical data is well supported by the structural data. All in all this is an interesting paper for the general reader. I particularly like how the structural data informs and guides consecutive biological experiments. However, there are a number of points, I would like the authors to address.

It is not clear if the second Mn²⁺ binding site has any biological relevance. This could easily be an artefact of protein preparation and soaking. The Mn²⁺ concentration during crystallisation and soaking is likely to be orders of magnitude higher than in the cell. The quantum mechanical calculations with a tiny subset of the protein taken out of structural context are not meaningful and do not really underline the biological argument of the paper.

The authors stress the point that this is 'druggable' target, however, the structure appears to be very similar to the human orthologue. This should be supported by genetic or chemical data (from the literature). For example, are there any antimicrobial compounds targeting this or closely related enzymes? What exactly is the consequence of 'weakened cell walls' as indicated in Figure 1 on the infections?

The crystal structures are generally very well determined but I could not help noticing that the two crystal forms are very similar, one having a doubled cell axis which leads to a non-crystallographic two-fold symmetry and leading to a non-conventional space description (P 2 2(1) 2(1)). Although this is not a crystallographic journal, a short discussion in the suppl. Material would be very much appreciated but the aficionado.

Line 87

Suggest to explain SRS repeats, and the other domains for the non-structural biologist. The depiction of CST1 in Figure 1a does not really correspond to the description in the text. In fact, I find the figure overcomplicated, enzyme in the figure have different names/abbreviations than in the text and in the figure caption. It remains unclear what "core 5"

Line 95

Suggest to explain 'GT-A fold' to the non specialist. What is the function of the lectin-type domain ?

Line 100

It would be useful to get an idea of the overall sequence similarity in the family of five enzymes mentioned.

Line 108

State what the sequence identities are between the T. Gondii and H. Sapiens, overall as well as in the active site. Only exact numbers give the reader a notion about the prospects of developing specific inhibitors.

Line 150

Suggest to explain what 'fully glycosylated' means ?

Line 179

Suggest to give rmsd between the Tg enzymes and the human homologue. I would also suggest to include the comparison, ideally a least-squares superposition at least in the suppl. Material, preferably in the main text.

Line 185

It would be useful to refer to average B-factors of ligand and compare these values with average B-factors of surrounding residues to provide confidence in the peptide site. All electron densities in the paper are only shown as (biased) 2Fo-Fc or Fo-Fc maps. It would have been much better to show unbiased Fo-Fc maps, or at least s.a. Fo-Fc maps.

Line 200

The discovery of the second metal binding site is intriguing, the anomalous data should be shown in the main manuscript. The wavelength/energy used for data collection needs to be added. Ideally, diffraction data should have been collected above or below the absorption edge of Mn.

Line 285

While it is plausible that the hydrophobic tail increases protein stability, the authors could have looked into its effect using (simple) TSA or DSF experiments.

Reviewer #3 (Remarks to the Author):

In humans and other warm blooded animals, an infection by *Toxoplasma gondii* persists for life in the form of latent cysts. The so-called bradyzoite cyst is surrounded by a wall whose function depends on appropriate O-glycosylation of some of its component proteins including Cst1. O-glycosylation of the Thr/Ser-rich mucin type domains of Cst1 is evidently initiated by any of three

polypeptide alpha-GalNAc transferases. This paper examines T.gondii-GalNAc-T3, which has an interesting dependence on the prior action of T.gondii-GalNAc-T1 or T2 in order to catalyze the addition of its own GalNAc to adjacent Thr/Ser residues. This and other features of T.gondii-GalNAc-T3 suggest that it might be different enough from its host polypeptide GalNAcTs to be a future drug target, thereby providing a good rationale for describing its structure and function in greater detail.

The well-done enzymatic, structural, and mutational studies of T.gondii-GalNAc-T3 reported here provide important information regarding its mode of action and reveal interesting novel features. A comprehensive set of peptide and glycopeptide substrates confirmed T3's preference to glycosylate a residue immediately N-terminal to a previously glycosylated residue. Remarkably, the X-ray crystal structures revealed an adaptation to the metazoan active site that involves a pocket with contribution of amino acids from both the catalytic and lectin-like domains, and an unusual second metal (Mn⁺⁺ here) binding site that binds the neighboring GlcNAc-O-Thr. The importance of this pocket was validated by kinetic analysis of point mutations of critical residues, and furthermore a correlation of kinetic and structure (experimental and computational) dependence on pH was examined to address mechanism. While glycosylation dependent enzymes occur in host cells, the mechanism of T3 is novel. The X-ray structures verified a functional Loop II involved in acceptor substrate binding, and revealed other novel features including an essential C-terminal extension (from the partially conserved C-terminal tri-lectin like domain) that reaches back to the catalytic domain. The functional significance of these features were also validated by point mutations in vitro. Importantly, many point mutants were also analysed in cyst walls of parasites differentiated into bradyzoite cysts and with regard to an expressed truncated Cst1 construct. Overall, the findings are comprehensive and well-documented, the in vivo comparisons provided important validation, and the evolutionary differences are intriguing. However, some aspects of interpretations detract from the final presentation and should be reviewed as indicated below.

Substantive issues

1. Line 330. Fig. S7. The experiment suggests an interesting dependence of the expressed Cst1 polypeptide on T3. The result is difficult to interpret without explanation of the expected MW of the polypeptide, presence of a loading control (e.g., why is the full-length band so faint?), and validation of glycosylation status using available anti-glycan reagents to confirm glycosylation status inferred from the rather dramatic band shifts.
2. Lines 351, 464. Statements are made that T3 acts processively. Yet elsewhere the mechanism is interpreted to be distributive. Evidence appears to support the latter. These statements need to be reconciled.
3. Line 366. The idea that Ca⁺⁺ potentially regulates T3 should address the differential compartmentalization of the enzyme and Ca⁺⁺ pools that are known to be regulated during the parasite-host interaction.
4. Line 384. I found it confusing to understand how the proposed SN2 mechanism resulting from the proximity of Glu332 would lead to a retention of configuration of the anomeric C1 of GalNAc. A clarification of the stereochemistry of GalNAc in Fig. 6B and more extended discussion is needed to clarify this model.
5. Line 82. The statement that polypeptide alpha-GalNAcTs are expressed throughout eukaryotes is not supported by evidence or the reference 5 cited. Other evidence indicates that, outside of animals, the only organisms that express polypeptide alpha-GalNAcTs rather than the highly conserved evolutionary predecessor, polypeptide alpha-GlcNAcTs, are apicomplexans that include T. gondii. This and the suggestion of lateral gene transfer that bears on the comparison of T3 with metazoan enzymes is reviewed in Mol Cell Proteomics (2021) 20, 100024.

6. Line 104. There is no direct evidence for a Core-5 structure. The assertion that the O-linked disaccharide is a Core-5 structure should be qualified as, e.g., described elsewhere (J Biol Chem (2019) 294, 1104)

Minor issues

7. Line 264. The designation Mn²⁺(B) is presumably a typo. But it also appears in Fig. S2, so bring all into conformity.

8. Line 452. Fig. 1b,c. Clarify to the reader that the assay in panel b directly measures transfer of GalNAc to peptide, whereas panel c indirectly measures activity by sugar nucleotide hydrolysis.

9. Line 452. Fig. 1g. ³H elution downstream of T* at Lys positions is presumably carryover during Edman sequencing. This should be acknowledged for clarity.

10. Line 613. 0.088 mM?

11. Line 646. What are the sources of Cst1.4, and Srs13.2? From Anaspec as implied?

12. Line 648. Source of UDP-GalNAc-F?

13. Line 673. In the kinetic assays reported here, sugar nucleotide hydrolysis commonly occurs in the absence of acceptor substrate for glycosyltransferases. Confirm that this does not occur for this enzyme or, if it did, was the value subtracted?

REVIEWER COMMENTS

Reviewer #1 (Remarks to the Author):

This study describes the substrate specificity and structure of a *Toxoplasma gondii* GalNAc-transferase-3 that is required for O-glycosylation of proteins in the bradyzoite cyst wall. Enzymes involved in the assembly of the heavily glycosylated cyst cell wall are considered potential drug targets for latent parasite stages. Initial studies confirmed that recombinant TgGalNAc-T3 transfers GalNAc to Thr residues in singly/multiply glycosylated peptides. GalNAc-T3 crystal structures in complex with an analogue of UDP-GalNAc and substrate glycopeptides revealed that the TgGalNAc-T enzyme has the same GT-A fold and conserved active site residues as mammalian GalNAcTs. Some differences were observed between the *Toxoplasma*/Apicomplexan GalNAc-Ts and mammalian GalNAc-Ts, including the presence of a second Mn²⁺ binding site (which may help binding to GalNAc), a surface exposed Loop II (which may modulate substrate binding) and a C-terminal tail (which binds to the catalytic domain). The functional importance of these domains in catalysis was supported by targeted mutagenesis studies. Finally, introduction of some of these mutations into the endogenous *T. gondii* GalNAc-T gene resulted in reduced cell wall O-glycosylation of in vitro-differentiated bradyzoite cysts, as well as reduced O-glycosylation of a secreted reporter protein, although no effect on bradyzoite differentiation was observed. This is a very nicely written and technically proficient study which represents a significant advance in our understanding of the structure and catalytic mechanism of this family of glycosyltransferase. The novelty of the findings are somewhat reduced by the previous studies which have already highlighted the importance of these enzymes for *T. gondii* cyst formation, and the lack of evidence that GalNAc-T3 is indeed druggable either in vitro or in vivo. In particular, the lack of a strong phenotype on bradyzoite differentiation following mutagenesis of key GalNAc-T3 residues and loss of cyst O-glycosylation suggests that any inhibitor will have to result in complete inhibition - a high bar that may not be achievable.

Major comments

Is metal binding site specific for Mn²⁺, or can it be substituted with Mg²⁺ or Ca²⁺?
Have the authors measured enzyme activity following removal of all cations and sequential substitution with different cations?

Thank you for the comment. It has been established that GalNAc-Ts are specifically Mn²⁺ dependent enzymes, where Mn²⁺ binds to the active site residues (including 2 Histidine, which are typically found bound to Mn²⁺ and not Mg²⁺), UDP, and a water. This has been shown for metazoan homologues and we have confirmed this preference for TgxGalNAc-T3 (Now included in Fig. S7). The standard TgxGalNAc-T3 buffer contains 0.5 mM EDTA, which would remove bound metal acquired during expression and purification and all assays are conducted using enzyme stored in EDTA containing buffer. We apologize for not including this step in the methods and have now incorporated buffer exchange into EDTA prior to assays/crystallization.

As far as the second site is concerned, we have not been able to decouple Mn^{2+}_N metal binding from the active site metal Mn^{2+}_A (although we have made a few attempts). Please see reasoning stated in response to reviewer 2. As stated in the discussion, it is possible that the Mn^{2+}_N site could bind other metals in vivo, such as Ca^{2+} . From our experience thus far, it would take a significant effort to determine conditions and could involve extensive additional mutagenesis to determine the “correct” metal for the second site, which is beyond the scope of this study but of interest to us for future work.

Fig 7. It is difficult to assess the extent to which mutations in GalNAc-T3 lead to reduced cyst wall glycosylation based on the images provided, as staining with the GalNAc-CST Ab is already weak in WT (bottom 20% of signal sensitivity). Can the authors quantitate GalNAc-CST1 fluorescence in the different bradyzoite preparations?

Thank you for the comment. We performed another experiment to quantify the cyst wall glycosylation. The cysts were stained using our polyclonal anti-CST1 antibody to identify the location of the cyst wall and stained using our anti-GalNAc-CST1 antibody to quantify the amount of GalNAc signal on the cyst wall. More than 2000 images were analyzed and quantified (see Figure 5 revised). This clearer approach demonstrates that point mutation of H333A resulted in an intermediate glycosylation phenotype, but that all other point mutants eliminated GalNAc glycosylation on the cyst wall.

Fig S7 is an important result and should be in the main text. However, the authors should also provide some indication of expression levels of the different mutated forms of GalNAc-T3. For example, do the mutations lead to loss of activity due to protein misfolding and clearance or due to reduced enzyme activity of equally expressing proteins.

*We have moved Fig S7 to the main text (Fig. 5b) and provided a loading control for this figure. We do not have reagents (antibodies) for *T. gondii* GalNAc-T3 which would allow quantification of the amount of protein in the various mutant strains. In construction of the mutant *T. gondii* strains a point mutation approach was used and, therefore, the strains do not have an HA or other tag that would allow quantification of the amount of protein or allow purification of the protein expressed in *T. gondii* to confirm the reduced enzyme activity demonstrated to in vitro for these mutants.*

How was differentiation performed (**currently no reference**)? The standard bradyzoite differentiation protocol utilizes high pH, which the authors have shown also results in loss of GalNAc-T3 activity. Is it possible that mutated proteins are just less stable than the WT protein at alkaline pH and inactivated by the harsh differentiation conditions?

Our paper on the identification of CST1 contains these conditions. The parasite within the host cell (within a parasitophorous vacuole) maintains a normal pH (pH7). The low CO₂ pH 8.2 culture conditions are a stress condition that trigger differentiation (as does many other stress conditions). The WT parasite has normal glycosylation and has the same exposure as the mutants.

Discussion -line 351– the authors suggest that TgGalNAc-T3 can act in a processive mechanism to successively add GalNAc residues to a peptide. It is not clear what they mean by processive. When applied to GTs, it normally means that the substrate remains bound to the enzyme while different sugars are added. However, this is contrary to what the authors proposed based on their substrate-specificity trials (Line 159).

Thank you for noticing. This was written by accident. The mechanism is distributive, not processive. We have edited the error.

There are no inhibitors for GalNAc-Ts at the moment despite significant interest and a recent study has suggested that it may be difficult to identify inhibitors of GalNAc-T1 (Nature Comms PMID: 36804936). This should perhaps be discussed.

*Indeed, there is significant interest in finding inhibitors of human GalNAc-Ts given their role in cancer and other diseases, but it has been quite difficult. However, there are inhibitors of GalNAc-Ts, including human GalNAc-T3 (Song, L. & Linstedt, A. D. Inhibitor of ppGalNAc-T3-mediated O-glycosylation blocks cancer cell invasiveness and lowers FGF23 levels. Elife 6, e24051 (2017)). One issue is that there are 20 human isoenzymes, and it is challenging to find selective inhibitors for each individual enzyme without effecting others. There is also little structural information available for many isoenzymes, including GalNAc-T11. The other challenge mentioned in the paper you cite is the lack of mammalian in vivo screening systems for the inhibitors. With improvements in bioengineering, there are now glycoengineered cell models for screening that have enabled the discovery of an inhibitor for human GalNAc-T3. Hopefully, these cell models can be used for further discovery. We have a few advantages for TxgGalNAc-T3 over the human isoenzymes that can be exploited. The structures and supporting data in this paper show that there are distinct, non-conserved regions in this enzyme that can be specifically targeted using structure-based design of inhibitors, which we also plan on doing and testing in *T. gondii*, which is a much simpler O-glycosylation system than the mammalian cell lines. While it is difficult to design inhibitors, it is not impossible.*

Reviewer #2 (Remarks to the Author):

This is a very good paper with a number of interesting structures. Overall the paper contains a lot of novel information and the biochemical data is well supported by the structural data. All in all this is an interesting paper for the general reader. I particularly like how the structural data informs and guides consecutive biological experiments. However, there are a number of points, I would like the authors to address.

It is not clear if the second Mn²⁺ binding site has any biological relevance. This could easily be an artefact of protein preparation and soaking. The Mn²⁺ concentration during crystallization and soaking is likely to be orders of magnitude higher than in the cell.

Thank you for the comment. We considered the possibility that this is an artefact of high concentrations of Mn^{2+} . However, the mutagenesis experiments were intended to address this point. In the initial manuscript, we show that changes to all the residues that bind the second metal (Mn^{2+N}) to Ala decrease enzymatic activity in vitro (Fig. 2d) and reduce O-glycosylation in vivo (Fig. 5). These residues also bind Thr-O-GalNAc, apart from E336, which only interacts with the second metal in the structure. Thus, we made additional variants of E336 that could be less disruptive to the enzyme fold, including E336Q and still see a decrease in activity. The lack of activity for E336Q supports a role for Mn^{2+N} in enzymatic activity and a biological role for the metal (Now included in Fig. 2d). For His333, we predicted that H333N would bind GalNAc, but not the metal. For that variant, we see a decrease in the K_M and k_{cat} , suggesting that the enzyme can still bind substrate, but cannot readily turnover the reaction, further supporting the role of metal in catalysis. Our computational studies additionally support a role for H333 in catalysis.

Having said that, the exact metal that binds Mn^{2+N} in vivo is a subject of our ongoing study that is beyond the scope of this manuscript. We are attempting to decouple the two metal binding sites in our activity assays. So far, despite several attempts, we have not yet found a way to do this. We made several attempts using Mn and Ca, because as we mention in the discussion, it is quite possible that this is a Ca binding site. We also tried the experiments with Mg^{2+} even though Mn^{2+N} is unlikely a Mg binding site given the presence of His and the coordination is not a perfect octahedral. We hope to provide more details on the role of the second metal site in a follow up studies.

The quantum mechanical calculations with a tiny subset of the protein taken out of structural context are not meaningful and do not really underline the biological argument of the paper.

Thank you for the comment, but we respectively disagree. We conducted quantum chemical calculations to examine the effect of the protonation state of His333 on the local geometry around the second Mn^{2+} . By analyzing the calculated local geometry, we provided a plausible rationale for the pH-dependent binding of GalNAc rather than making hand-waving statements. It is worth noting that Reviewer #3 complemented our approach, saying "...furthermore a correlation of kinetic and structure (experimental and computational) dependence on pH was examined to address mechanism."

*Currently, we are carrying out computer simulations with the hybrid potentials of QM/MM to investigate the mechanism of *T. gondii*-GalNAc-T3. These QM/MM simulations are more complex and challenging. They may also better assess the effect of the protonation state of His333 on the binding of GalNAc. However, these simulations are beyond the scope of the present work.*

The authors stress the point that this is 'druggable' target, however, the structure appears to be very similar to the human orthologue. This should be supported by genetic or chemical data (from the literature). For example, are there any antimicrobial compounds targeting this or closely related enzymes? What exactly is the

consequence of 'weakened cell walls' as indicated in Figure 1 on the infections ?

TxgGalNAc-T3 is similar to the human isoenzymes in many ways. However, the distinct (and potentially druggable) regions in TxgGalNAc-T3 are the ones we emphasize throughout the manuscript and summarize in Fig. 6a and 6b. Nevertheless, we deemphasized the druggability of this enzyme in the updated version of the manuscript because the goal of this paper was to characterize these enzymes structurally so that we had information that could be used in the future to design inhibitors.

Since GalNAc-Ts are predominantly present in higher eukaryotes, and this manuscript describes the first characterization of an isoenzyme from a pathogen, there are no existing anti-microbials. However, there has been a significant effort to find inhibitors of these enzymes in human disease. For instance, inhibitors for GalNAc-T3 and GalNAc-T11 mentioned by reviewer 1 (Nature Comms PMID: 36804936).

In response to the comment regarding the weakened cyst wall, the idea is that bradyzoites have thus far been resistant to drugs. Since the cyst wall is critical for latent infection, weakening the cyst wall may allow the immune system to clear the cysts and latent infection, and moreover prevent reactivation to tachyzoites. Additionally, because cyst walls may prevent drugs from getting into the cyst and targeting the bradyzoite, weakening the cyst wall could allow access to drugs in combination therapy. We have updated the discussion with this commentary.

The crystal structures are generally very well determined but I could not help noticing that the two crystal forms are very similar, one having a doubled cell axis which leads to a non-crystallographic two-fold symmetry and leading to a non-conventional space description (P 2 2(1) 2(1)). Although this is not a crystallographic journal, a short discussion in the suppl. Material would be very much appreciated but the aficionado.

Thank you for the feedback. We processed and scaled each data set in multiple orthorhombic space groups and the way that h, k, and l were assigned by the software HKL2000 for 6 of the structures, resulted in the assigned group being P22121 independent of which scaled data is used and which molecular replacement approach is taken (ie whether you use Phaser or MolRep (CCP4)). These structures contain 1 molecule/asymmetric unit, and this has now been updated in the crystallographic tables.

What we are not certain about is the structures in the space group P212121. The commonality between these 2 structures is that they both contain a peptide with GalNAc at the N-terminus (Muc5Ac-3 and SRS13.2, Table S1) and have two molecules in the asymmetric unit (updated in table S1). Given the lack of electron density for the peptides, it is not clear why the asymmetric unit symmetry is not "broken" in these structures, but it is indeed interesting.

We have included a version of this commentary in supplemental results.

Line 87

Suggest to explain SRS repeats, and the other domains for the non-structural biologist. The depiction of CST1 in Figure 1a does not really correspond to the description in the text. In fact, I find the figure overcomplicated, enzyme in the figure have different names/abbreviations than in the text and in the figure caption. It remains unclear what “core 5”.

Thank you for the suggestions.

SRS repeats: SRS (SAG1 related repeats) repeats refer to a structure (PMID: 15003490) first identified in surface antigen 1 (SAG1/p30) that is found in a superfamily of about 160 proteins in T. gondii. The exact function of these SRS domains is not known, but they are thought to mediate protein interactions, perhaps with host cell surface proteins.

We expanded the description which now states: It contains thirteen SAG1 related sequence (SRS) domains found in a superfamily of T. gondii surface antigens that could facilitate parasite entry into host cells.

CST1 depiction: It is not clear how the figure and text are different. Both the text and Fig. 1A describe the SRS repeats, mucin domain and C-terminal Cys rich region similarly.

Thank you for the comment on Fig. 1. We simplified by fixing names/abbreviations and removed the core5 label.

Line 95

Suggest to explain ‘GT-A fold’ to the non-specialist. What is the function of the lectin-type domain?

Thank you for the suggestion. We tried to keep it simple but appreciate that the lack of details is somewhat confusing. Thus, we updated the text to describe the GT-A fold more extensively as a Rossman-like fold.

We hope that this clarifies the description of this family of enzymes.

Line 100

It would be useful to get an idea of the overall sequence similarity in the family of five enzymes mentioned.

Thank you for the comment. We included a % identity in the text, a percent identity matrix in Fig. S1, and mapped out sequence similarity on T_{xg}GalNAc-T3 in Fig. S4c.

Line 108

State what the sequence identities are between the T. Gondii and H. Sapiens, overall as well as in the active site. Only exact numbers give the reader a notion about the

prospects of developing specific inhibitors.

Thank you for the comment. While these numbers provide useful information, inhibitors are about folded 3D structure for these enzymes. Even a few non-conserved amino acids can influence binding pockets and substrate specificity of each enzyme, particularly in this enzyme family where differences in substrate specificity are encoded within non-conserved domains. Nevertheless, we have included sequence identity data (from Clustal Omega) in the text and included an identity matrix in Fig. S1.

The sequence identity for the catalytic domain is similar, so we used sequence similarity to give more insight into how this structure compares to the human isoenzymes. We made a sequence alignment using Clustal Omega within ChimeraX and show how sequence similarity, which is less conservative than sequence identity, mapped out onto the structure of TxgGalNAc-T3. The sequence similarity assessment shows that the active site is more conserved than surrounding regions and the lectin domain. We have included a figure to illustrate sequence conservation in Fig. S4c.

Either way, we are not focused on the active site in this manuscript as a drug target, other than E332 which is not highly conserved, and spend a significant effort discussing variable sequences in regions of interest throughout the manuscript. In the sequence similarity figure, the second metal site is not conserved, the C-terminal tail is unique to TXGGalNAc-T3, and both loops 1 and 2 are highly variable.

Line 150

Suggest to explain what 'fully glycosylated' means?

Thank you for the comment. We changed fully for "highly" glycosylated and described that as most sites being modified with GalNAc.

Line 179

Suggest to give rmsd between the Tg enzymes and the human homologue. I would also suggest to include the comparison, ideally a least-squares superposition at least in the suppl. Material, preferably in the main text.

Thank you for the suggestion. We have calculated RMSD in ChimeraX and show the comparisons on the TxgGalNAc-T3 structure in Fig. S4b.

Line 185

It would be useful to refer to average B-factors of ligand and compare these values with average B-factors of surrounding residues to provide confidence in the peptide site. All electron densities in the paper are only shown as (biased) 2Fo-Fc or Fo-Fc maps. It would have been much better to show unbiased Fo-Fc maps, or at least s.a. Fo-Fc maps.

The peptide site is highly conserved among GalNAc-Ts. Nevertheless, we have included a B-factor colored structure to show the B-factors of surrounding residues for

both CST1.4 and Muc5AC3,13 structures. These analyses are shown in Fig. S5a and b. In addition, we have calculated the unbiased Fo-Fc maps for the active site and peptides, and these are shown in the Fig S5c and d.

Line 200

The discovery of the second metal binding site is intriguing, the anomalous data should be shown in the main manuscript. The wavelength/energy used for data collection needs to be added. Ideally, diffraction data should have been collected above or below the absorption edge of Mn.

Thank you for the suggestion. We have moved the anomalous data to the main text and included the energy of data collection (1 Å) which is below the absorption edge of Mn (1.8 Å) in Tables S1 and S4. Unfortunately the beam line was reluctant to change the energy to 1.4 or 1.5 Å due to potential issues with instability for other users. Nevertheless, we see an anomalous signal for both active site and 2nd site Mn ions.

Line 285

While it is plausible that the hydrophobic tail increases proteins stability, the authors could have looked into its effect using (simple) TSA or DSF experiments.

Thank you for the suggestion. We have conducted SEC analysis and DSF for the T₃GalNAc-T3 variants generated. These are all shown in Fig. S6 and S8. In addition, we have referred to the data in the text.

Reviewer #3 (Remarks to the Author):

In humans and other warm-blooded animals, an infection by *Toxoplasma gondii* persists for life in the form of latent cysts. The so-called bradyzoite cyst is surrounded by a wall whose function depends on appropriate O-glycosylation of some of its component proteins including Cst1. O-glycosylation of the Thr/Ser-rich mucin type domains of Cst1 is evidently initiated by any of three polypeptide α—GalNAc transferases. This paper examines T.*gondii*-GalNAc-T3, which has an interesting dependence on the prior action of T.*gondii*-GalNAc-T1 or T2 in order to catalyze the addition of its own GalNAc to adjacent Thr/Ser residues. This and other features of T.*gondii*-GalNAc-T3 suggest that it might be different enough from its host polypeptide GalNAcTs to be a future drug target, thereby providing a good rationale for describing its structure and function in greater detail.

The well-done enzymatic, structural, and mutational studies of T.*gondii*-GalNAc-T3 reported here provide important information regarding its mode of action and reveal interesting novel features. A comprehensive set of peptide and glycopeptide substrates confirmed T3's preference to glycosylate a residue immediately N-terminal to a previously glycosylated residue. Remarkably, the X-ray crystal structures revealed an adaptation to the metazoan active site that involves a pocket with contribution of amino acids from both the catalytic and lectin-like domains, and an unusual second metal (Mn⁺⁺ here) binding site that binds the neighboring GlcNAc-O-Thr. The importance of

this pocket was validated by kinetic analysis of point mutations of critical residues, and furthermore a correlation of kinetic and structure (experimental and computational) dependence on pH was examined to address mechanism. While glycosylation dependent enzymes occur in host cells, the mechanism of T3 is novel. The X-ray structures verified a functional Loop II involved in acceptor substrate binding and revealed other novel features including an essential C-terminal extension (from the partially conserved C-terminal tri-lectin like domain) that reaches back to the catalytic domain. The functional significance of these features were also validated by point mutations in vitro. Importantly, many point mutants were also analysed in cyst walls of parasites differentiated into bradyzoite cysts and with regard to an expressed truncated Cst1 construct. Overall, the findings are comprehensive and well-documented, the in vivo comparisons provided important validation, and the evolutionary differences are intriguing. However, some aspects of interpretations detract from the final presentation and should be reviewed as indicated below.

Substantive issues

1. Line 330. Fig. S7. The experiment suggests an interesting dependence of the expressed Cst1 polypeptide on T3. The result is difficult to interpret without explanation of the expected MW of the polypeptide, presence of a loading control (e.g., why is the full-length band so faint?), and validation of glycosylation status using available anti-glycan reagents to confirm glycosylation status inferred from the rather dramatic band shifts.

The expected unmodified molecular weight of surrogate mucin protein is 46.1 kDa. The observed high molecular weight band does not represent the full-length CST1 with SRS domains; it corresponds only to the expressed mucin domain. The band is faint since the conditions for high molecular weight protein transfer to obtain transfer of the high molecular weight protein efficiently while keeping the low molecular weight proteins transferred are challenging. We use conditions for transfer that allow us to identify both the high and low molecular weight proteins but agree that quantification of the various bands has significant experimental contingencies. As suggested, we provided a loading control using a T. gondii specific dense granule protein 1 (GRA1) antibody. This result was incorporated into the figure for publication. Our prior publications provide clear data that the anti-GalNAc-CST1 antibody (an IgE monoclonal antibody) binds to the mucin domain of CST1 and that this domain is glycosylated in CST1. New data to address this comment are all present in Fig,3

2. Lines 351, 464. Statements are made that T3 acts processively. Yet elsewhere the mechanism is interpreted to be distributive. Evidence appears to support the latter. These statements need to be reconciled.

Thank you for noticing. You are correct that the mechanism is most likely distributive. The text has been clarified.

3. Line 366. The idea that Ca⁺⁺ potentially regulates T3 should address the differential

compartmentalization of the enzyme and Ca⁺⁺ pools that are known to be regulated during the parasite-host interaction.

Thank you for the comment. We have updated the discussion and commented on the role of the Golgi as a calcium storage organelle, although it does not seem that there is a lot known about how the calcium leaks into the cytosol during the tachyzoite lytic cycle.

4. Line 384. I found it confusing to understand how the proposed SN2 mechanism resulting from the proximity of Glu332 would lead to a retention of configuration of the anomeric C1 of GalNAc. A clarification of the stereochemistry of GalNAc in Fig. 6B and more extended discussion is needed to clarify this model.

*Thank you for the comment. An SN2 reaction would result in an inversion of configuration. From Edman degradation, we know that the GalNAc is alpha-linked to the peptide. The issue here is for that to happen, the GalNAc would have to be Beta-linked to the UDP. However, there is no evidence that I could find in the literature to suggest that this is the case, and UDP-GlcNAc and UDP-GalNAc biosynthesis results in a product with alpha-linkage. That it isn't to say that it is not possible that *T. gondii* has an enzyme that makes Beta linked UDP-GalNAc. Thus, SN2 is possible, but unless there is evidence of Beta-linked UDP-GalNAc in *TGondii*, it would be difficult .*

Alternatively (and more likely), the reaction mechanism could be double displacement, resulting in retention of configuration, which fits better with the idea that both UDP-GalNAc and Thr-O-GalNAc are alpha-linked. E332, which is currently ~5.5 Ang away from C1, approaches the C1 for a nucleophilic attack, which is followed by nucleophilic attack by the acceptor Thr for double displacement. E332 is on the same loop as H333, which binds metal. It is possible that at high pH, tight metal binding restricts loop movement and prevents E332 from attacking the C1 carbon (E332 is adjacent to H333). When metal binding is weaker (pH 7), the loop has more flexibility and E332 can "move" 2 Ang for a nucleophilic attack. We have updated Fig 6 accordingly to focus on the more plausible mechanism.

Whether the reaction is SN2 or double displacement is a subject of future investigation. We are carrying out computer simulations with the hybrid potentials of QM/MM. These studies will also provide further insight into the inhibition of this enzyme, by perhaps targeting intermediate or transition states during catalysis.

5. Line 82. The statement that polypeptide alpha-GalNAcTs are expressed throughout eukaryotes is not supported by evidence or the reference 5 cited. Other evidence indicates that, outside of animals, the only organisms that express polypeptide alpha-GalNAcTs rather than the highly conserved evolutionary predecessor, polypeptide alpha-GlcNAcTs, are apicomplexans that include *T. gondii*. This and the suggestion of lateral gene transfer that bears on the comparison of T3 with metazoan enzymes is reviewed in Mol Cell Proteomics (2021) 20, 100024.

Thank you for the comment and reference. The text has been updated and now reads "The cyst wall contains a subset of proteins that undergo mucin-type O-glycosylation, a post-translational modification (PTM) that occurs on proteins that pass through the secretory pathway and is conserved across higher eukaryotes and a subset of apicomplexan protozoa". You are correct that the review you suggested accurately reflects the description we provided. Thank you for sharing the review for this paper, but it is also generally a fantastic source of information.

6. Line 104. There is no direct evidence for a Core-5 structure. The assertion that the O-linked disaccharide is a Core-5 structure should be qualified as, e.g., described elsewhere (J Biol Chem (2019) 294, 1104)

We apologize for the missing reference that was the source for our statement about Core-5 structures, this was an oversight during editing. We have cited the above paper and toned down the conclusiveness of the structure being Core 5.

Minor issues:

7. Line 264. The designation Mn²⁺(B) is presumably a typo. But it also appears in Fig. S2, so bring all into conformity.

Thank you for noticing. We have changed incidents of Mn²⁺_B to Mn²⁺_N.

8. Line 452. Fig. 1b,c. Clarify to the reader that the assay in panel b directly measures transfer of GalNAc to peptide, whereas panel c indirectly measures activity by sugar nucleotide hydrolysis.

Thank you for the comment. We have included these clarifications in the legend for Fig. 1.

9. Line 452. Fig. 1g. 3H elution downstream of T* at Lys positions is presumably carryover during Edman sequencing. This should be acknowledged for clarity.

We have included a sentence at the end of the legend to Fig 1 noting the sequencer lag for the 3H-GalNAc-O-Thr-PTH derivatives and a similar comment to the legend to supplemental figure S1 where the lag is also observed.

10. Line 613. 0.088 mM?

Thank you for noting this error, which has been corrected

11. Line 646. What are the sources of Cst1.4, and Srs13.2? From Anaspec as implied?

Yes, these were from AnaSpec while others are from Sussex Research. We have therefore added the sources of all the peptides and glycopeptides used in each section of the methods.

12. Line 648. Source of UDP-GalNAc-F?

Thank you, we have updated the text. UDP-GalNAc-F was synthesized by ChemBind (<http://chembind.com>, Atlanta GA).

13. Line 673. In the kinetic assays reported here, sugar nucleotide hydrolysis commonly occurs in the absence of acceptor substrate for glycosyltransferases. Confirm that this does not occur for this enzyme or, if it did, was the value subtracted?

Thank you for this comment. In our reactions of T_xgGalNAc-T₃ against the CST1.x and SRS13.x and Muc5AC.x glycopeptides we noted that there was essentially no observed hydrolysis of UDP-GalNAc as demonstrated from our G10 analysis (Fig. S11). We have therefore noted this in the methods section discussing the sequencing of these glycopeptide substrates as well as in the methods section describing the UDP-Glo assay. We have also included a comment here that we do see UDP-GalNAc hydrolysis for glycopeptide substrates that lack a directly N-terminal acceptor relative to a glycosylated Thr.

REVIEWERS' COMMENTS

Reviewer #1 (Remarks to the Author):

I thank the authors for addressing the major comments raised in the reviews. The revised manuscript is improved and the study overall makes a significant contribution to our understanding of the glycobiology of these parasites.

Reviewer #2 (Remarks to the Author):

I had a close look at the revised manuscript and it is clear that the authors have taken (almost) all suggestions by the reviewers into account. In my view this is an important piece of work suitable for Nature Communication and hence I support publication.

Reviewer #3 (Remarks to the Author):

The revision is substantially improved with new documentation and improved interpretations of existing and new data. All of my previous concerns have been satisfactorily addressed.

Importantly, the previously proposed SN2 catalytic mechanism has been revisited. The currently proposed double displacement mechanism makes more sense with new data regarding E332. However, until recently this was an unprecedented mechanism for retaining glycosyltransferases, which is not acknowledged. Interestingly, new evidence is emerging in support of a double displacement mechanism for two other CAZy GT families:
<https://doi.org/10.1016/j.jbc.2023.105006>

Regarding the comment that not much is known regarding regulation of cytosolic Ca⁺⁺ in tachzoites, the authors might consider updating reference 36 with a recent article from the same group:
<https://doi.org/10.1016/j.jbc.2024.105771>

Paragraph starting at line 442, instances of Glu322 should be Glu332.

REVIEWERS' COMMENTS

Reviewer #1 (Remarks to the Author):

I thank the authors for addressing the major comments raised in the reviews. The revised manuscript is improved, and the study overall makes a significant contribution to our understanding of the glycobiology of these parasites.

Thank you, we appreciate your time and feedback.

Reviewer #2 (Remarks to the Author):

I had a close look at the revised manuscript and it is clear that the authors have taken (almost) all suggestions by the reviewers into account. In my view this is an important piece of work suitable for Nature Communication and hence I support publication.

Thank you, we appreciate your time and feedback.

Reviewer #3 (Remarks to the Author):

The revision is substantially improved with new documentation and improved interpretations of existing and new data. All of my previous concerns have been satisfactorily addressed.

Importantly, the previously proposed SN2 catalytic mechanism has been revisited. The currently proposed double displacement mechanism makes more sense with new data regarding E332. However, until recently this was an unprecedented mechanism for retaining glycosyltransferases, which is not acknowledged. Interestingly, new evidence is emerging in support of a double displacement mechanism for two other CAZy GT families: <https://doi.org/10.1016/j.jbc.2023.105006>

Thank you for sharing the review from Dr. Guerin. That is quite interesting, and we clearly have a lot more to learn about GTs. I have included the two original research references in the manuscript.

Regarding the comment that not much is known regarding regulation of cytosolic Ca⁺⁺ in tachzoites, the authors might consider updating reference 36 with a recent article from the same group: <https://doi.org/10.1016/j.jbc.2024.105771>

Thank you for sharing this new research from the group further describing calcium entry in T. gondii, which we have included as a reference. However, given the recent work and the new discoveries, I took out the sentence “not much is known...”.

Paragraph starting at line 442, instances of Glu322 should be Glu332.

Thank you for noticing. They have been fixed.

Finally, thank you to all the reviewers for taking the time to read and review the manuscript. Your constructive comments helped us to improve the quality of the manuscript, and we appreciate your enthusiasm.